# A supramolecular gel-elastomer system for soft iontronic adhesives

Dace Gao [1,7], Gurunathan Thangavel[1,5,7], Junwoo Lee[2,6,7], Jian Lv [1,3], Yi Li[4], Jing-Hao Ciou[1], Jiaqing Xiong [1], Taiho Park [2] & Pooi See Lee [1,3] ✉

Electroadhesion provides a promising route to augment robotic functionalities with continuous, astrictive, and reversible adhesion force. However, the lack of suitable conductive/dielectric materials and processing capabilities have impeded the integration of electroadhesive modules into soft robots requiring both mechanical compliance and robustness. We present herein an iontronic adhesive based on a dynamically crosslinked gel-elastomer system, including an ionic organohydrogel as adhesive electrodes and a resilient polyurethane with high electrostatic energy density as dielectric layers. Through supramolecular design and synthesis, the dual-material system exhibits cohesive heterolayer bonding and autonomous self-healing from damages. Iontronic soft grippers that seamlessly integrate actuation, adhesive prehension, and exteroceptive sensation are devised via additive manufacturing. The grippers can capture soft and deformable items, bear high payload under reduced voltage input, and rapidly release foreign objects in contrast to electroadhesives. Our materials and iontronic mechanisms pave the way for future advancement in adhesive-enhanced multifunctional soft devices.

Soft robots empowered with controllable adhesives can benefit from astrictive, tunable, and reversible interfacial attraction to achieve unconventional robotic functionalities. Bioinspired gecko-adhesion[1–3] employs micro- or nanoscopic artificial setae array to produce Van der Waals interaction for universal adhesion, yet it is mechanically complicated to mimic a gecko's toe-pad motion to adhere and detach these adhesives. Electroadhesion (Fig. 1a) offers an alternative approach that is mechanically simple, lightweight, and easily switchable between engagement and release, thus have been utilized in recent robotic prototypes for anti-gravity locomotion[4–6], aerial perching[7], and fragile object handling[8–12] (Table 1). However, incorporating electroadhesion into soft robotic bodies remains challenging. Existing device-level

electroadhesives are commonly operated under a few kilovolts to generate sufficient adhesive force, and have limited materials selection to fabricate with. Electroadhesives made out of flexible contact layer (e.g., polyimide) and metallic thin film electrodes[8,13] can generate robust electroadhesion on flat, smooth surfaces, but fail to accommodate free-form or rough surfaces, and lack the elasticity to integrate with fully soft-bodied robots. Electroadhesives based on elastomeric components[9], such as silicone and stretchable electronic conductor (e.g., carbon-filler-percolated composite) feature improved geometrical adaptation and conductor-dielectric integration[9,14], but are limited by the low dielectric constant of silicones, as well as the irreversible loss in adhesive functionality upon severed damages (e.g.,

[1]School of Materials Science and Engineering, Nanyang Technological University, 50 Nanyang Avenue, Singapore 639798, Singapore. [2]Department of Chemical Engineering, Pohang University of Science and Technology, Pohang 37673, Republic of Korea. [3]Singapore-HUJ Alliance for Research and Enterprise (SHARE), Smart Grippers for Soft Robotics (SGSR), Campus for Research Excellence and Technological Enterprise (CREATE), Singapore 138602, Singapore. [4]School of Electrical Engineering and Automation, Wuhan University, Wuhan 430072, China. [5]Present address: Advanced Materials Research Center, Technology Innovation Institute (TII), Masdar City, Abu Dhabi P.O Box 9639, United Arab Emirates. [6]Present address: Department of Chemical and Environmental Engineering, Yale University, New Haven, CT 06511, USA. [7]These authors contributed equally: Dace Gao, Gurunathan Thangavel, Junwoo Lee. ✉ e-mail: pslee@ntu.edu.sg

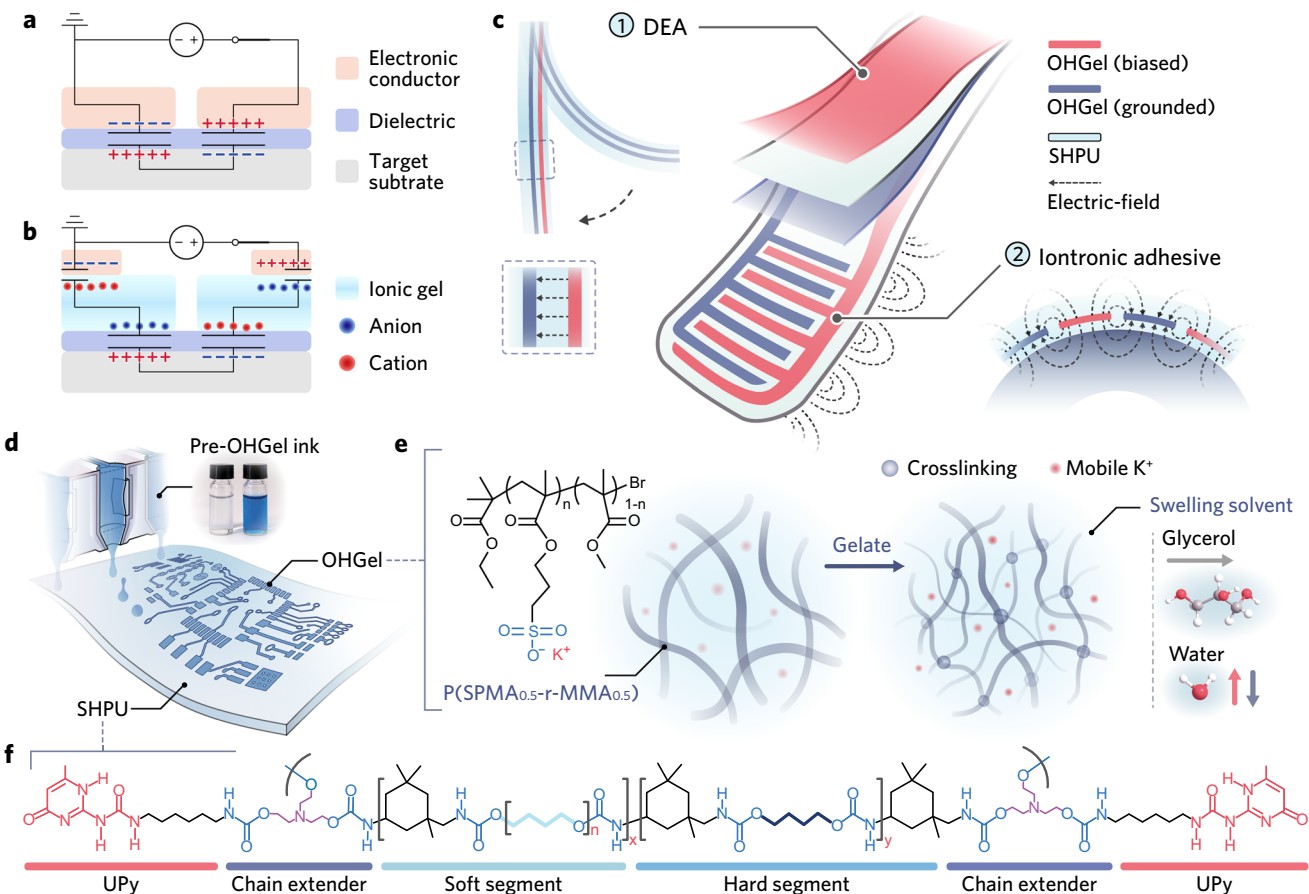

**Fig. 1 | Electrostatic adhesion mechanisms and supramolecular OHGel-SHPU system. a**, **b** Schematic illustrations and equivalent circuits of (**a**) electroadhesives and (**b**) iontronic adhesives when adhering on a conductive substrate. **c** Conceptual illustrations of a soft iontronic adhesive unit, and the mechanisms of DEA and electrostatic adhesion. The exact device structural design is available in Fig. 4a. **d** Illustrative inkjet printing process of OHGel electrodes on a SHPU

substrate. OHGel is inherently transparent and could be colored with toluidine blue to enhance its visibility. **e** Macromolecular structure of P(SPMA$_{0.5}$-r-MMA$_{0.5}$, left), the gelation process of pre-OHGel ink after inkjet printing (middle), and the composition of the swelling solvent (right; oxygen, carbon, and hydrogen atoms are colored red, grey, and white, respectively). **f** Macromolecular structure of SHPU, $n = 15$, x:y = 0.65:0.35.

**Table 1 | Comparison of state-of-the-art electrostatic adhesive devices**

| Application | Materials & fabrication | | | Performance | | |
| --- | --- | --- | --- | --- | --- | --- |
| | Electrode | Contact layer | Patterning method | Operating voltage (kV) | Normal pressure (kPa), substrate | Shear pressure (kPa), substrate |
| Aerial perching[7] | Cu | PI | Sputtering with mask | 1.0 | 0.20, glass 0.21, steel | NA |
| Wall climbing[4] | Cu | PI | Sputtering with mask | 5.0 | 0.35, glass 0.51, paper | 1.05, glass 3.43, paper |
| Wall climbing[6] | Carbon grease | Silicone | NA | 4.0 | NA | 4.1, glass 2.4, paper |
| Wafer handling[8] | Cu | PI | Sputtering with mask | 2.0 | 1.5, glass 1.6, Al | NA |
| Gripper[12] | Au | PI | Evaporation with mask | 3.0 | 0.7, Ge-coated PI | 5.0, Ge-coated PI |
| Gripper[10] | Cu | Silicone | Chemical etching | 5.0 | 4.8, acrylic elastomer | 5.6, acrylic elastomer |
| Gripper[9] | Carbon composite | Silicone | Stamp printing | 5.0 | 13, PMMA* | 35, PMMA* |
| Gripper[11] | Carbon composite | Clarifoil film | Laser cutting | 4.8 | NA | 0.64, PMMA |
| Gripper (this work) | Ionic OHGel | SHPU | Inkjet printing | 1.0 | 1.3, glass 2.1, Al | 2.4, glass 4.7, Al |

*PI* Polyimide, *PMMA* Polymethyl methacrylate. *A paper coating is applied on the acrylic to avoid dry adhesion (Van der Waals force).

cutting, tearing, and puncturing). Therefore, further advancements of electrostatic adhesives will demand new classes of soft conductors and dielectric elastomers with favorable material property and processibility to manifest strong electrostatic adhesion force under reduced voltage input.

Soft iontronics[15] are emerging devices that employ ion-conducting gels as readily stretchable electrodes. Unlike electronic conductors that tend to loss conductive paths under large strain, ionic

gels can maintain ion transportation (with slight increment in resistivity)[16] even under hyperelastic deformation[17–19]. Moreover, ionic gels of sophisticated macromolecular design can manifest remarkable cycle stability, fatigue resistance[20], self-healability[21,22], and compatibility with additive fabrication[23]. By establishing non-faradaic electrical double layers (EDLs) between ionic gel electrodes and peripheral electronics (e.g., processors and power source), and coupling ionic gels with electroactive/passive dielectric elastomers, various soft

iontronic devices including artificial muscles[16], proprioceptive[24]/exteroceptive[25,26] sensors, and electroluminescent light emitters[19,27] were developed with profound implications for multifunctional, intelligent soft robots.

In this article, we report an ionic gel-enabled electrostatic adhesion configuration (Fig. 1b) and the delivery of iontronic adhesives based on a supramolecular gel-elastomer system. By virtue of the unique material characteristics and the ionic-electronic hybridized framework, our iontronic adhesives exhibit strong electrostatic adhesion at reduced voltages ($\leq 1\,kV$, see Table 1 for comparison with electroadhesives), fast release from foreign surfaces ascribing to the transient discharge of EDL capacitors, and autonomous self-healability under ambient conditions. Being designed, synthesized (Supplementary Figs. 1–3), and assembled using noncovalent crosslinking in a bottom-up fashion, the ionic organohydrogel (OHGel) and the supramolecular, hierarchically H-bonded polyurethane (SHPU) are successfully empowered by dynamic chain association to restore its electromechanical properties from severed damages, and to form robust OHGel-SHPU interfacial bonding through ion-dipole interactions. We present a millimeter-scale ($9 \times 32\,mm^2$) gripping unit—a dielectric elastomer actuator (DEA) unimorph with a seamlessly integrated iontronic-adhesive "fingertip" (Fig. 1c, see Supplementary Fig. 4 for detailed working mechanisms)—by inkjet printing OHGel electrodes and layering up SHPU membranes in a designated sequence (see Supplementary Methods for details) without using bonding agents. The assembly of four gripping units produces a soft gripper that benefits from astrictive adhesion to manipulate fragile items, meanwhile yields strong shear holding forces and consequently exceptional payload-to-weight ratios (e.g., 670 on metal) thanks to the dielectric/tribological properties of SHPU and the micropatterning of OHGel electrodes. Furthermore, we demonstrate a self-healable iontronic-adhesive patch whose device integrity and functionality can recover from a complete cut across a SHPU-OHGel-SHPU multilayered region. This work herein lays a solid platform for fully self-restorable iontronic soft robots, in which distributed ionic gel electrodes along with dielectric matrices can self-heal concomitantly to improve robotic reliability and sustainability over the long run.

## Results

### Supramolecular OHGel-SHPU system

We developed ionic OHGel as electrode material in favor of its inherent compliance, healability, and compatibility with high-resolution additive patterning processes. We previously reported the synthesis of $P(SPMA_{0.5}\text{-}r\text{-}MMA_{0.5})$[28], an amphiphilic polyelectrolyte that spontaneously gelate in polar solvents. Here we modified the polymer's synthetic route (Supplementary Methods) to regulate its polydispersity (Supplementary Fig. 5) so as to render the $P(SPMA_{0.5}\text{-}r\text{-}MMA_{0.5})$-based pre-OHGel ink suitable for inkjet printing (Fig. 1d, see printability analysis in Supplementary Fig. 6). When blended with a water-glycerol binary solvent, the polymer's ionic side chains hydrate whilst its hydrophobic blocks desolvate and self-assembly into a supramolecular network through hydrophobic interaction[29], producing the physically crosslinked OHGel upon partial dehydration (Fig. 1e). The incorporation of nonvolatile, hygroscopic glycerol helps to retain water and no sign of gel stiffening was observed over the course of this study. An increasing glycerol-polyelectrolyte ratio preserves more initial water in OHGel (Supplementary Fig. 7) and thereby softens OHGel due to the enhanced degree of polyelectrolyte-solvent interaction (see Fourier transform infrared spectroscopy (FTIR) results, Supplementary Fig. 8). Consequently, the elastic modulus ($E$) and ultimate tensile strain ($\varepsilon_u$) of OHGel are facilely tunable over orders of magnitude (~70 kPa to 3500 kPa and ~280% to ~2000%, respectively, Fig. 2a) to cater for different robotic applications. The optimal OHGel for our iontronic-adhesive grippers has a

polyelectrolyte-glycerol ratio of 2:1 (w/w, denoted as PE10/GY5, referred to as OHGel in the following context), which gives rise to a balanced deformability ($E \approx 360$ kPa, $\varepsilon_u \approx 800\%$), elasticity (higher storage modulus than loss modulus, Supplementary Fig. 9), and self-healing efficiency (100% mechanical strength recovery within 15 min, Supplementary Fig. 10). Moreover, OHGel exhibits a high ionic conductivity at room temperature ($4.92 \times 10^{-3}$ S cm$^{-1}$) based on the transportation of mobile potassium ions. Owing to the anti-freezing/drying nature (freezing point $\approx -43$ °C, boiling point $\approx 111$ °C) of the water-glycerol mixture (-60:40, w/w)[30], OHGel maintains conductive from $-20$ to 80 °C and exhibits a temperature-activated conducting behavior following the Arrhenius law[31] (Fig. 2b and Supplementary Fig. 11).

In our soft iontronic devices, SHPU (Fig. 1f) serves as an ideal carrier and insulative encapsulation material that protects OHGel electrodes from mechanical damages such as accidental perforation (Fig. 2c). SHPU is a microphase-separated thermoplastic elastomer (TPE, schematic illustration in Fig. 2d) that unites seemingly antagonistic properties: it possesses the kinetic reversibility to self-heal at room temperature, yet exhibits felicitous toughness that imparts robustness to soft robots. Specifically, SHPU comprises semicrystalline hard domains (H-bonded urethane aggregation, $T_g \approx 65$ °C) that render high tensile strength ($\sigma_u \approx 14.4$ MPa), and an amorphous soft matrix ($T_g \approx -12$ °C) that gives rise to large stretchability ($\varepsilon_u \approx 2000\%$). The large divergence in glass transition temperatures ($T_g$) suggests a prominent immiscibility between the hard and soft phases, leading to superior toughness (100.8 MJ m$^{-3}$, Fig. 2e, red curve) in SHPU bulk elastomer. Whereas classical TPEs exploit covalent connectivity to join the soft matrix, we incorporated ureidopyrimidinone (UPy)[32] telechelic groups as quadruple H-bonding motifs to dynamically associate the soft phase, and further harness the dynamic chain motion in soft matrix to realize reversible UPy-UPy dimerization for self-healing. Details for the design, synthesis, and characterization of SHPU macromolecule are available in Supplementary Methods, Supplementary Fig. 12-13 and Supplementary Table 1. Microphase separation in SHPU was evidenced by thermogravimetric analysis (TGA, Supplementary Fig. 14a) and dynamic mechanical analysis (DMA, Supplementary Fig. 14b). The morphology of hard domains was further investigated via small-angle X-ray scattering (SAXS), where the intensive and broad scattering peaks detected from $-20$ to 80 °C (Supplementary Fig. 14c) indicate the presence of thermally stable, well-defined nanospheres with an average inter-domain spacing of ~5 nm. UPy dimers are dispersed in the soft matrix to assist self-healing as no profile of UPy π-π stacking[33] (another type of hard domain) was observed in the SAXS results.

While the viscoelastic character of many self-healable elastomers precludes their usage in soft robotics, SHPU features low-hysteresis elasticity as the high association constant[34] of UPy dimers ($K = 6 \times 10^8$ M$^{-1}$) can effectively suppress plastic deformation in the soft matrix. Upon consecutive stretch-release cycling, SHPU displayed 20.1% and 10.4% energy dissipation in the first and tenth loop, respectively (Fig. 2f), which contrasts distinctly with the pronounced elastic energy loss (>50%) of previously reported self-healable elastomers[35–37]. After resting for 15 min following the initial cycle, SHPU could recover from viscoelastic deformation as suggested by the almost identical stress-strain loops (Supplementary Fig. 15). In terms of electrical property, the enriched dipoles in SHPU render a higher dielectric constant ($\kappa \approx 6.8$, 100 Hz) than commercial acrylic (VHB 4905) and silicone elastomers (Sylgard 184, Fig. 2g), allowing the material to render a same level of electrostatic energy density ($\kappa\varepsilon_0 E^2$, $\varepsilon_0$ is vacuum permittivity, $E$ is the nominal electric field across a dielectric layer) with reduced voltage input. Moreover, the breakdown field ($E_b$) of SHPU was measured to be 63.6 V μm$^{-1}$ through Weibull analysis (Supplementary Fig. 16a, b), and its maximum electrostatic energy density[38] ($\kappa\varepsilon_0 E_b^2$) was calculated to be 0.243 MJ m$^{-3}$. A thorough comparison in electrical properties

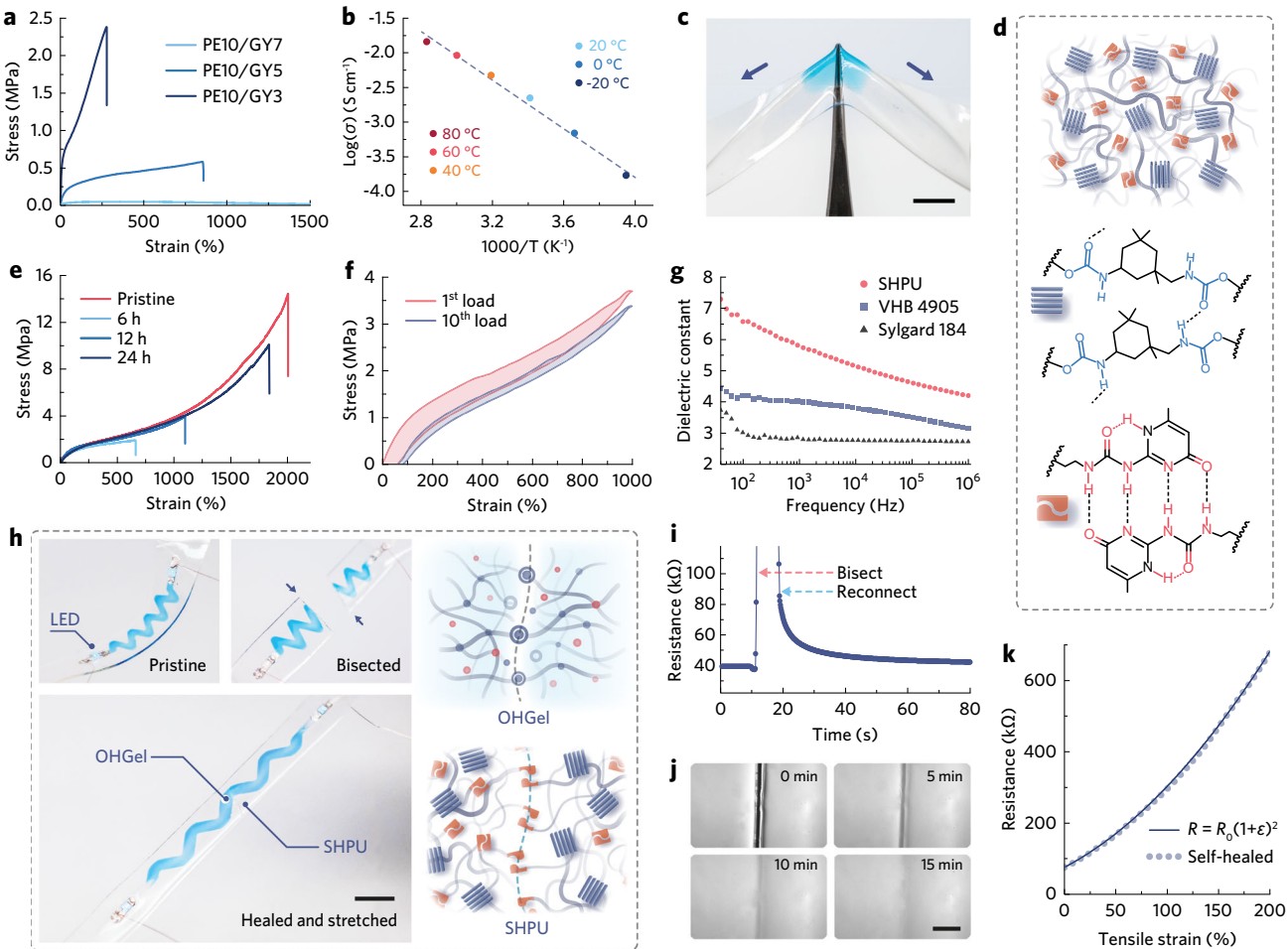

**Fig. 2 | Electromechanical properties of OHGel and SHPU. a** Uniaxial tensile test results of OHGels with varying PE/GY ratio. Stretching speed, 60 mm min⁻¹. **b** Temperature dependence of OHGel's ionic conductivity from −20 °C to 80 °C. **c** Photograph showing a SHPU substrate with OHGel loaded on top resisting the puncturing from a sharp tweezer. Scale bar, 5 mm. **d** Schematic illustration representing the hard-soft phase separation in SHPU (top); molecular structures of urethane hard domain (middle) and UPy-UPy dimer (bottom) that represent the hierarchical H-bonding in SHPU. **e** Stress-strain behaviors of pristine and self-healed SHPU samples after different healing durations. **f** Stress-strain cyclic behavior of SHPU under successive tensile loading up to 1000% strain. **g** Dielectric constant of SHPU, VHB 4905, and Sylgard 184 as a function of sampling frequency from 40 Hz to 10⁶ Hz. **h** Photographs recording the self-healing process of an OHGel-SHPU composite (left, scale bar, 10 mm); Schematic illustrations of the self-healing mechanisms in OHGel and SHPU (right). **i** Resistance change of the OHGel electrode upon bisection and reconnection. **j** Optical microscopic images recording the mechanical self-healing process of OHGel. Scale bar, 100 μm. **k** Resistance change of the OHGel electrode under uniaxial tensile strain after it was fully self-healed from damage.

between SHPU, VHB, and Sylgard 184 are provided in Supplementary Fig. 16c. Note that $E_b$ is not an intrinsic material property, but a measured parameter that depends on sample geometries and testing conditions. The herein claimed values are valid when referring to our testing protocol (elaborated below Supplementary Fig. 16).

As demonstrated in Fig. 2h, we cut across an OHGel-SHPU interconnect, i.e., an OHGel electrode printed on a SHPU substrate and interconnecting two terminal light emitting diodes (LEDs), then rejoined the bisections to investigate the self-healing efficiency of our supramolecular material system. OHGel exhibited a two-step healing process, including an instantaneous recovery in ionic conductance (~90%, 22.9 out of 25.5 μS) within 40 s (Fig. 2i), and a subsequent reconstruction of dynamic crosslinks (reassociation of free hydrophobes) that fuse the cut interface over 15 min (Fig. 2j, Supplementary Fig. 10). Upon the mechanical self-healing of the SHPU substrate, the interconnect could be again stretched (~200% strain) while keeping the LEDs powered. During uniaxial tensile stretch, the normalized resistance ($R/R_0$) of the healed OHGel electrode increased as $(1+\varepsilon)^2$ (Fig. 2k, $\varepsilon$ is the tensile strain), confirming that the cut in OHGel was electromechanically healed. SHPU was self-restored under ambient

conditions ascribing to the rapid kinetics of UPy-UPy association (off-rate constant, $k_{off} \approx 8\,s^{-1}$). For dumbbell-shaped SHPU samples, the stress-strain curves obtained after bisecting and healing for different hours followed closely the pristine one (Fig. 2e). A 24 h healing period led to 91.8%, 70.1%, and 72.3% recovery in $\varepsilon_u$, $\sigma_u$, and toughness, respectively, which demonstrates a top-tier mechanical robustness and self-healing efficiency in dielectric elastomers as reported to date (see Ashby plot comparison in Supplementary Fig. 17 and Table 2).

## Advanced manufacturing of iontronic-adhesive grippers

The supramolecular OHGel-SHPU system enables us to develop iontronic soft robots with unparalleled advantages in additive fabrication and multi-material assembly, leading to improved integration level within a small footprint. Empowered by the superior inkjet printability of the pre-OHGel ink, planar and interdigitated OHGel electrodes can be facilely deposited onto SHPU membranes with ~30 μm feature resolution and a uniform thickness profile below 1 μm (see line printing results in Fig. 3a). Such an OHGel patterning protocol facilitates the rapid prototyping of ionic gel circuits (on elastomeric substrates) with sophisticated geometries (Fig. 3b), which

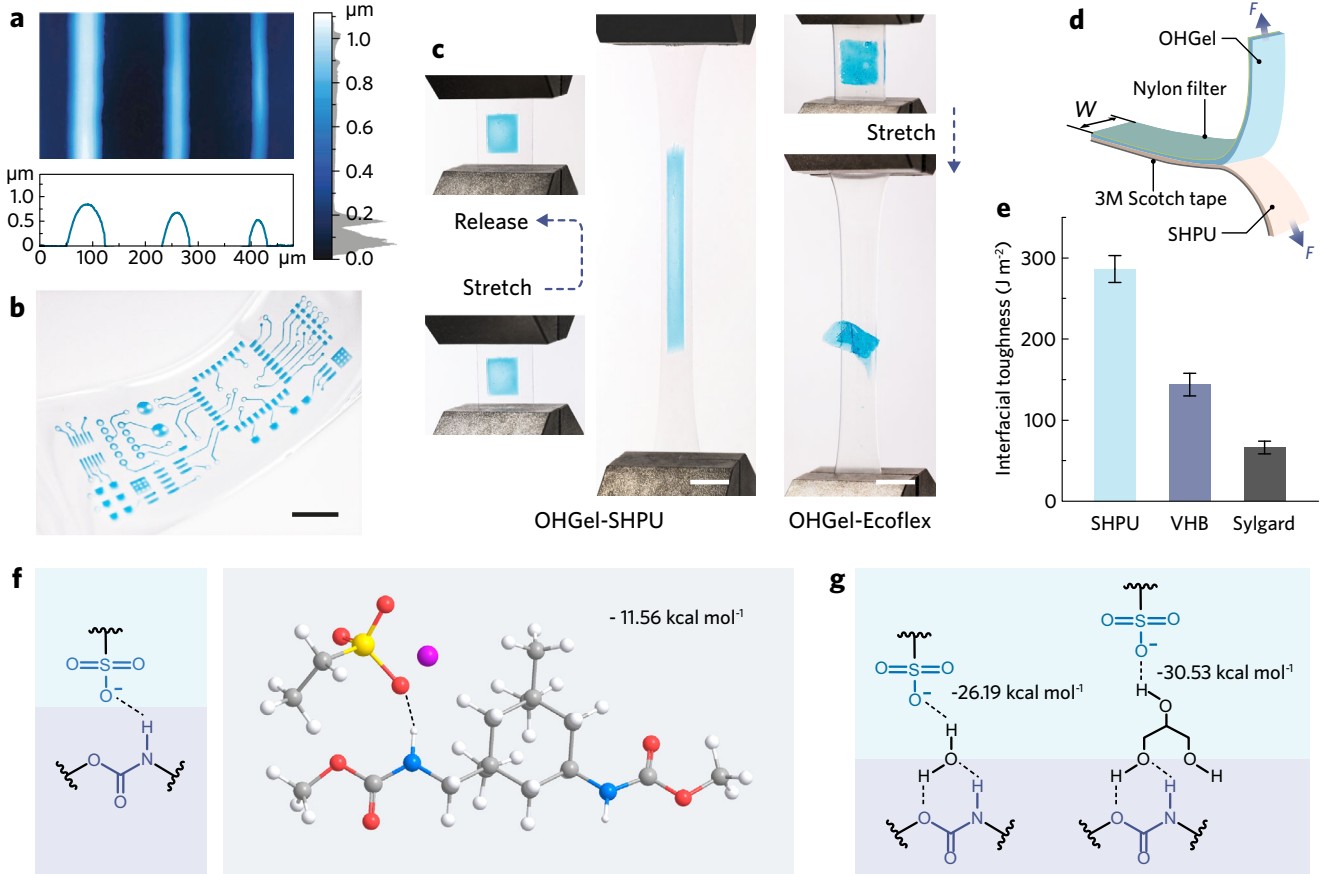

**Fig. 3 | Additive manufacturing and robust OHGel-SHPU interface. a** 3D-topography of inkjet-printed OHGel lines captured by confocal microscope. **b** Photograph of an inkjet-printed OHGel-SHPU ionic circuit with sophisticated electrode patterns. Scale bar, 5 mm. **c** Photographs showing an OHGel-SHPU composite under large tensile deformation ($\varepsilon > 600\%$) without debonding, and an OHGel-Ecoflex composite delaminating under much lower strain ($\varepsilon < 100\%$). Scale bars, 10 mm. **d** Schematic illustration of ASTM D1876 T-peeling test. **e** Measured interfacial toughness between OHGel and different elastomers (distinct samples, mean ± s.d., $n = 5$). **f** Chemical structure and DFT result depicting the ion-dipole interaction between urethane (in SHPU) and sulfonate (in OHGel) groups. **g** Chemical structures representing the water/glycerol bridged ion-dipole interactions.

potentially provides a high-resolution, programmable, and pattern-designable fabrication platform for iontronic applications[39,40]. Recently, Ge et al. reported a series of 3D-printed structures featuring covalent bonding between acrylamide hydrogel and acrylate elastomers[41]. Zhang et al. also reported a multi-material 3D printing technique that covalently bond ionic and dielectric elastomers[42]. In this work, OHGel-SHPU composite delivers an alternative gel-elastomer system that forms strong and inherent interfacial bonding without requiring additional surface treatment[43] or coupling agent[44]. As shown in Fig. 3c and Supplementary Video 1, an OHGel-SHPU bilayer could be cyclically stretched up to 600% strain without delamination, whereas OHGel loaded on Ecoflex started to detach at 100% strain. A high interfacial toughness of $286.4 \pm 16.6\,\text{J m}^{-2}$ was measured between OHGel and SHPU by T-peeling tests (ASTM D1876, Fig. 3d and Supplementary Fig. 18), which outperformed the bonding toughness between OHGel and acrylic/silicone elastomers (Fig. 3e). Density functional theory (DFT) calculations suggest that the strong cohesion at OHGel-SHPU interface can originate from ion-dipole interactions[22] (Fig. 3f), where an electropositive hydrogen in urethane group (in SHPU) interacts with a negatively charged sulfonate group (in OHGel) to form a heterogeneous ionic H-bond[45] (N-H···O = S, binding energy $\Delta E = -11.56\,\text{kcal mol}^{-1}$). In case when the sulfonate group is solvated, such interaction can be bridged through H-bonded water or glycerol molecules (Fig. 3g and Supplementary Fig. 19). We also performed Raman spectroscopy to elucidate the molecular events at the OHGel-SHPU interface, where the decreased

intensity in sulfonate vibration ($1046\,\text{cm}^{-1}$) suggests its association with exotic proton donors (Supplementary Fig. 20).

As such, the automated inkjet printing technique together with the self-bonding between OHGel-SHPU layers allowed us to rapidly assemble a soft yet robust gripping unit consisting of a dorsal DEA and an anterior end effector (Fig. 4a). The DEA adopts a dielectric elastomer minimum energy structure (DEMES)[46] of which the bending curvature at rest is approximated using Timoshenko analysis[47] (Supplementary Fig. 21) and is tunable by adjusting the prestretch in the active SHPU interlayer. The end effector, composed of interdigitated OHGel electrodes and a SHPU contact layer, can perform either electrostatic adhesion or capacitive sensing as the situation demands. Importantly, the contact layer employs a superhydrophobic coating (fluorinated silica nanoparticle, Supplementary Fig. 22a) on its outer surface to obviate the inherent tackiness of SHPU, which consequently eliminates post-gripping adherence[14] and enables self-cleaning by preventing the collection of contaminants (Supplementary Fig. 23). Finally, multiple gripping units can be assembled to complete an iontronic-adhesive gripper (Fig. 4b).

**Versatile object manipulation with ultrahigh payload and gentle touch**

To establish power supply and signal communication, OHGel electrodes are connected to electrical leads (peripheral circuits) by interfacing with a nanoporous carbon composite (Supplementary Fig. 22b), where electrical double layers (EDLs) form and behave like

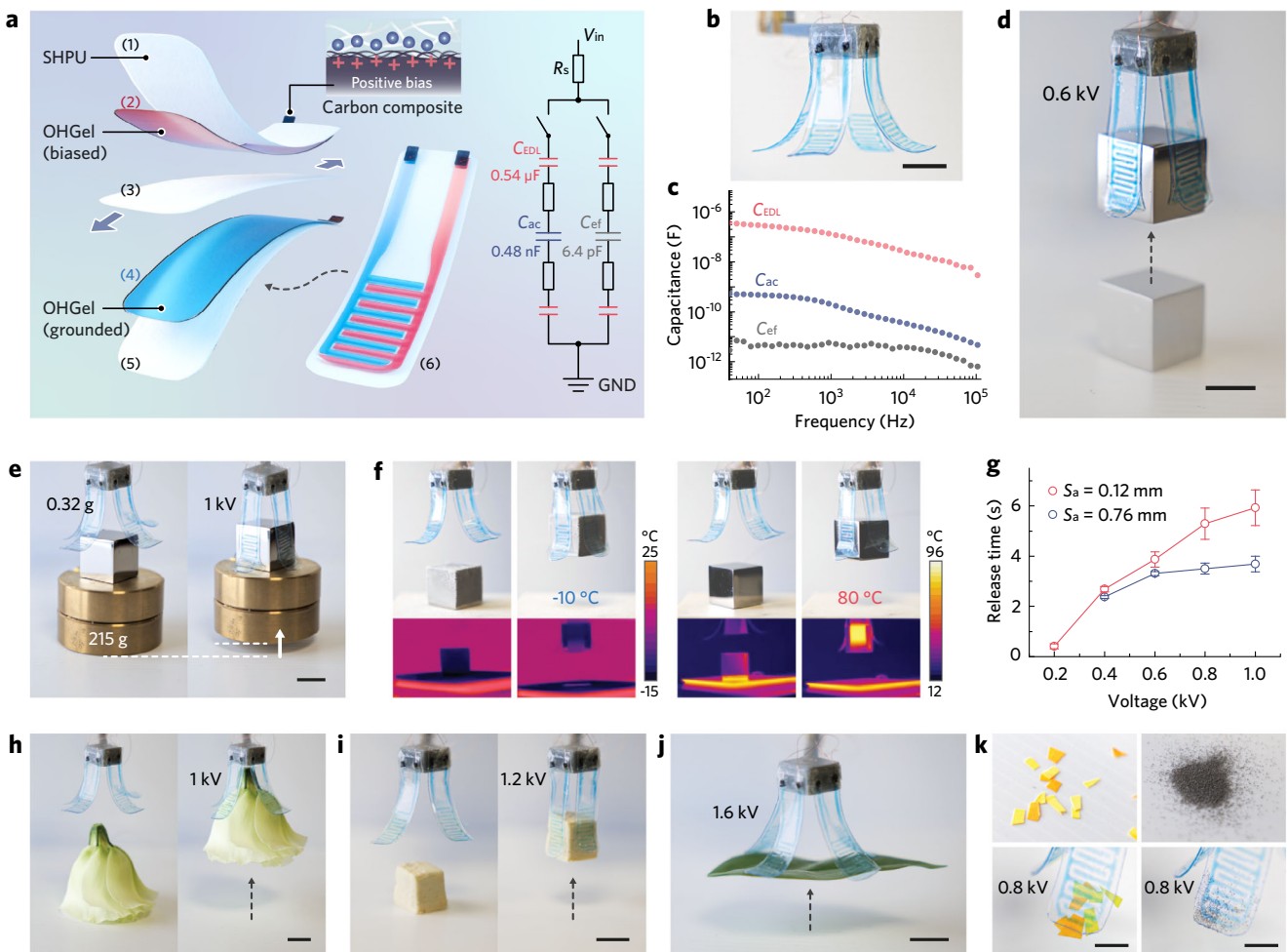

**Fig. 4 | Robust and versatile iontronic-adhesive gripper. a** Structural design of an iontronic-adhesive gripping unit consisting of OHGel electrodes and SHPU dielectric layers (left). In the DEA module, a pair of parallel-plate OHGel electrodes (2, 4) are separated by a prestretched SHPU membrane (3), then encapsulated by passive SHPU layers (1, 5) on both sides. The thickness of top (1), middle (3, pre-stretched), and bottom (5) SHPU layer is 60 μm, 60 μm, and 80 μm, respectively. The end effector module comprising interdigitated OHGel electrodes and a SHPU contact layer (6, thickness = 60 μm) is further attached below layer (5) to form the gripping unit. The inset diagram represents the equivalent circuit of the gripping unit, where $R_s$ denotes the series resistance induced by wiring, and the non-labelled resistors denote the ionic resistance of OHGel electrodes. **b** Photograph of an iontronic-adhesive soft gripper at rest. Scale bar, 10 mm. **c** Capacitance of $C_{EDL}$, $C_{ac}$, and $C_{ef}$ as a function of sampling frequency from 40 Hz to $10^5$ Hz. **d** Photograph showing the gripper picking up a metallic cube (15 g, titanium) under 0.6 kV voltage input. Scale bar, 10 mm. **e** Photographs showing the gripper lifting a 205 g object by harnessing the strong shear adhesion generated on metallic surfaces. Scale bar, 10 mm. **f** Photographs and thermographic images showing the gripper capturing metallic cubes of high/low temperatures. **g** Recorded time intervals that the gripper needed to release the metallic cube after turning off the voltage input (same sample measured repeatedly, mean ± s.d., $n = 5$). **h–k** The Iontronic-adhesive gripper demonstrating its versatility by picking up a flower (**h**), a piece of tofu (**i**), a flat leaf (**j**), and tiny objects such as paper shreds and titanium particles (**k**). Scale bars, **h–j** 10 mm; **k** 5 mm.

volumetric capacitors ($C_{EDL}$) in series connection with either the parallel-plate capacitor ($C_{ac}$) in the unimorph DEA or the coplanar capacitor ($C_{ef}$) in the end effector (see equivalent circuit in Fig. 4a inset). While $C_{EDL}$ is orders of magnitude larger than $C_{ac}$ and $C_{ef}$ (Fig. 4c), a voltage input would preferentially couple across the dielectric capacitors for electromechanical transduction without complications arising from electrochemical reactions at the EDLs[16]. Thereby, direct current (DC) supply will actuate the DEA beam meanwhile activate the end effector to polarize a nearby foreign object, resulting in electrostatic adhesion between the coplanar OHGel electrodes and the mating surface. Taking advantage of the high dielectric constant and low hysteresis of SHPU, the low thickness (60 μm) of SHPU contact layer, and the densely patterned OHGel segmentation (width = 0.9 mm, pitch = 0.4 mm), the gripper could be readily triggered by a 0.6 kV DC input to capture a metallic cube within 0.2 s (Fig. 4d and Supplementary Video 2), where the unimorph actuation initiated the gripping motion and the onset of

electrostatic attraction before physical contact accelerated the engagement. Although the iontronic gripper is lightweight (≈0.32 g, excluding the weight of the holder), it could uplift a 215 g metallic object under 1 kV (Fig. 4e) and thus demonstrate an ultrahigh payload-to-weight ratio (≈670, see Supplementary Table 3 for comparison). The overall robustness of our gripper stems from the high mechanical toughness in SHPU that protects the grippers from rupturing when holding heavy loads, and the strong OHGel-SHPU interlayer cohesion that prevents the multilayered device from delamination. The heat/freezing tolerance of the material system (demonstrated in Supplementary Fig. 24 and Supplementary Video 3) allowed the gripper to handle a metallic cube that is either ice-cold (−10 °C) or scorching (80 °C, Fig. 4f, Supplementary Video 4). Besides, iontronic adhesives feature rapid release from a conductive surface (Fig. 4g) in contrast to the prolonged detaching time (mins to hrs) for electroadhesives due to the retention of residue charges[48]. We presumably attribute the fast release to a synergy between the

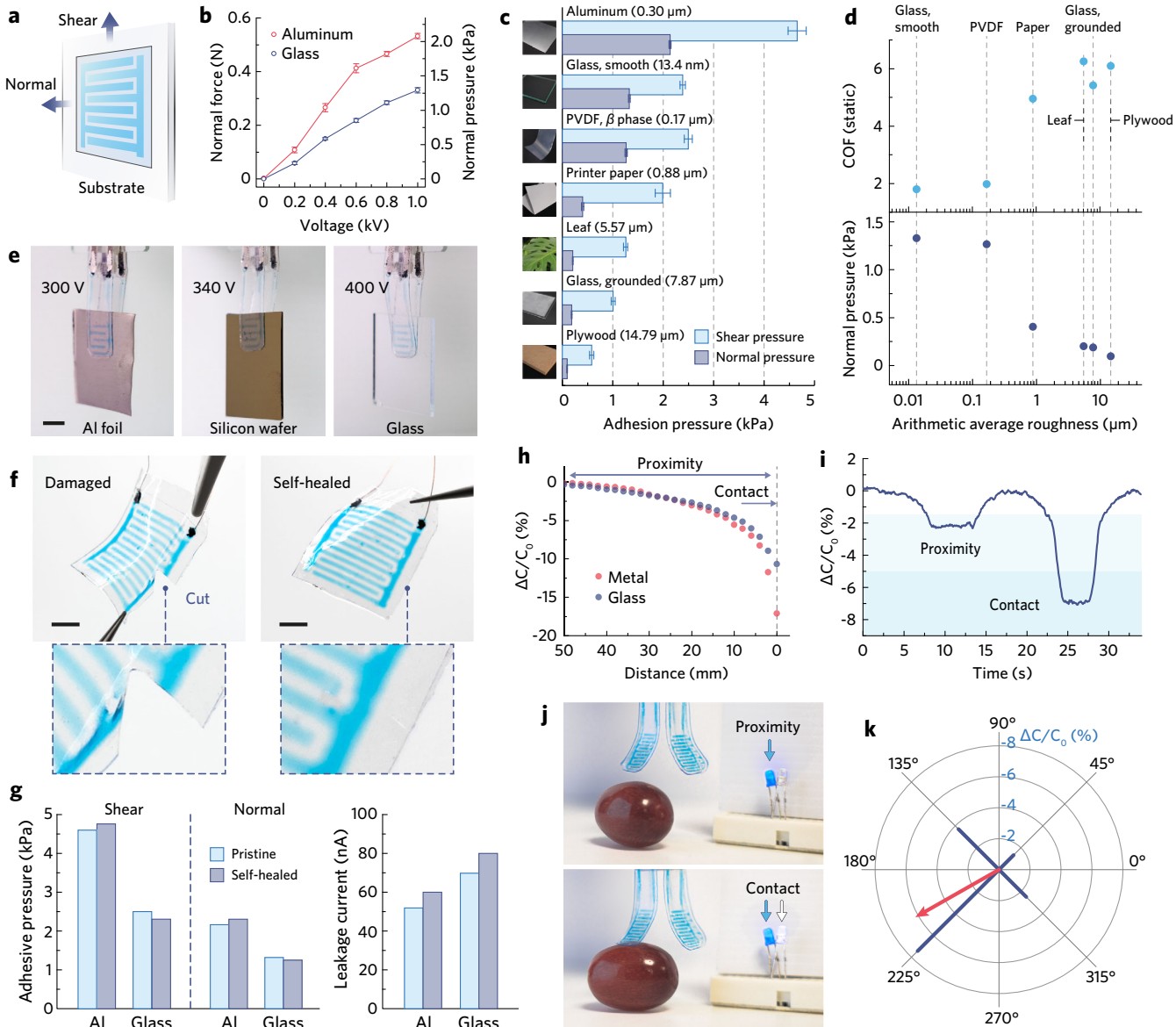

**Fig. 5 | Adhesion performance and exteroceptive sensation. a** Schematic illustration depicting the testing methods for normal and shear adhesion pressures. **b** Recorded normal adhesion force (and pressure) of iontronic-adhesive patch on aluminum/glass substrates under varying voltage inputs (same sample measured repeatedly, mean ± s.d., n = 5). **c** Recorded normal and shear adhesion pressure between the iontronic-adhesive patch and various substrates under 1 kV (same sample measured repeatedly, mean ± s.e., n = 5). **d** Analysis of the correlation between COF, normal adhesion, and $S_a$. **e** Photographs showing a two-fingered gripper picking up aluminum foil, silicon wafer, and smooth glass (from left to right) under low driving voltage. Scale bars, 5 mm. **f** Photographs showing an iontronic-adhesive patch before and after self-healing from a cut damage. Scale bars, 5 mm. **g** Column charts comparing the normal/shear adhesion pressure and leakage current of an iontronic-adhesive patch before and after self-healing (same sample measured repeatedly, mean, n = 5). **h** Capacitance change in an end effector as the gripper approached the target object beneath it. **i** Continuously recorded capacitance change showing that the end effector can differentiate if the target is in proximity or in contact. **j** Photographs showing the communication between the end effectors and the LED indicators. The LED setup can change its lighting pattern regarding to proximity or contact state. **k** Radar map showing that the gripper can measure capacitance changes from all the end effectors and resolve the object's location by adding up the vectors.

built-in (back-scrolling) stress in DEMES and the fast self-discharging[49] of $C_{EDL}$ that expedite charge redistribution in OHGel electrodes (Supplementary Fig. 25).

In addition to its strength in weightlifting, our iontronic-adhesive gripper also excels in handling delicate objects by exerting astrictive adhesion instead of localized compression (Supplementary Video 5). When picking up a soft and deformable flower, normal adhesion kept the end effectors in conformal contact with the petals, while shear adhesion yielded sufficient friction to lift the flower (Fig. 4h). Also demonstrated was the successful handling of a piece tofu (Fig. 4i) that is fragile, water-rich, and easily broken when being mechanically

compressed. By directly harnessing normal adhesion, the gripper could also adhere to flat surfaces and manipulate objects that lack grabbable features such as a piece of leaf, as well as tiny items like paper confetti and metallic microparticles (Fig. 4j, k, Supplementary Video 6).

## Adhesion analyses and device-level self-healability
We pursued a standardized testing method to quantify the adhesive pressure generated by our iontronic-adhesive devices (Fig. 5a, testing setups and data interpretation methods are available in Supplementary Fig. 26) and to understand how the normal and shear pressures are

correlated. Iontronic-adhesive patches of a specific electrode geometry (active electrode area = 16 × 16 mm, Supplementary Fig. 27) were fabricated, then tested on aluminum, glass (with varying surface roughness), polyvinylidene difluoride (PVDF) film, printer paper, leaf, and plywood, whose surface texture and average roughness ($S_a$) are given in Supplementary Fig. 28. With fixed geometrical parameters (electrodes pattern, contact SHPU layer thickness, etc.), an iontronic-adhesive patch measured higher normal pressure with increasing voltage input and manifested stronger effect on conductors (aluminum, 2.12 kPa under 1 kV) than dielectrics (glass, 1.33 kPa under 1 kV, Fig. 5b). $S_a$ is another critical parameter of our interest as it influences normal and shear adhesion in divergent trends. In a general description of static friction, shear pressure ($P_T$) is related to normal pressure ($P_N$) by $P_T = \mu_s P_N$, where $\mu_s$ denotes the static coefficient of friction (COF) between a SHPU layer and a substate in contact. Normal adhesive pressure is inversely related to surface asperities (due to more pronounced micro air gaps) and the patch-to-glass normal adhesion (under 1 kV) could drop over 7 folds (from 1.33 to 0.19 kPa) when $S_a$ of glass increased from 13.4 to 7.87 μm (Fig. 5c). On the other hand, a rougher solid surface can yield a higher COF (Fig. 5d) under static friction as is related to the microscopic deformation and delayed recovery (hysteresis) in SHPU contact layer[50], thus providing sufficient shear force for the gripping of rough-surfaced objects. For example, the shear adhesive pressure on grounded glass could maintain ~1 kPa despite the declined pressure in normal direction.

The above studies suggest that the iontronic-adhesive patch under 1 kV can generate substantially high shear pressure on metals (e.g., 4.66 kPa on aluminum) and smooth/high-k dielectrics (e.g., 2.39 kPa on glass, 2.50 kPa on PVDF), and exploitable shear pressure on rough/low-k dielectrics (e.g., 1.01 kPa on grounded glass, 0.58 kPa on plywood). Reducing the operating voltage down to 400 V still allows the patch to produce a shear pressure of 2.26, 1.28, and 0.50 kPa on aluminum, smooth glass, and grounded glass, respectively, which corresponds to a payload of 86.1 g, 50.6 g, and 19.8 g for our four-fingered iontronic-adhesive gripper. (Supplementary Fig. 29). The substantially reduced driving voltage enables the two-fingered gripper (Fig. 5e and Supplementary Video 7) to pick up a piece of aluminum foil, silicon wafer, and smooth glass under 300, 360, and 400 V, respectively. Voltage reduction also leads to less leakage current and power consumption (at microwatt level, Supplementary Fig. 30), and provides the feasibility to integrate low-mass power sources (<1 g)[51] into untethered grippers in future prototypes. Moreover, the development of supramolecular OHGel-SHPU material system enabled us to devise the first mechanically tough and self-healable iontronic-adhesive patch that recovered from a severed cut across OHGel and SHPU layers (Fig. 5f) and continued to adhere on various substrates. Self-healing in multilayered soft devices is considerably challenging because viscoelastic elastomers/gels tend to deform plastically under shear cutting, resulting in distorted edges that are difficult to realign. In contrast, the high toughness and elasticity in SHPU contribute to clean and less-deformed cut surfaces (Supplementary Fig. 31) so that both the OHGel electrode and the SHPU encapsulations can achieve good realignment with higher self-healing yield. The healed iontronic-adhesive patch exhibited comparable adhesion performance to pristine devices (Fig. 5g), only accompanied with slightly increased leakage current that may compromise its sustainability under high voltage for large force output.

### Exteroceptive sensation
We explored the gripper's exteroceptive sensing capability to allow probing the spatial location of a target object. In our iontronic-adhesive gripper, a charged end effector projects fringe electric field into its neighboring space and the field is prone to be disturbed by a nearby object, which in return results in a lower measurement in $C_{ef}$ (Fig. 5h). Capacitive sensing can thus be executed by reading out the relative change ($\Delta C/C_0$) in $C_{ef}$ using a capacitance-to-digital convertor

(CDC), then processing the signal with a microcontroller (see circuit design in Supplementary Fig. 32). As an example, the gripper could detect an object such as a grape and determine if it is in proximity ($\Delta C/C_0 = -2.4\%$) or in contact ($\Delta C/C_0 = -7.2\%$) by defining a threshold (Fig. 5i and Supplementary Video 8). Communicating the microcontroller with a set of peripheral LEDs further provided visual indications of the interacting state (Fig. 5j). When multiple end effectors functioned collaboratively, the divergence in capacitance change would enable the gripper to spatially resolve the object's position (Fig. 5k) and adjust the succedent gripping motion accordingly.

## Discussion
In this study, we presented that supramolecular iontronics and additive microfabrication could cooperatively transform the design and functionalization of soft robots. The as developed OHGel-SHPU material system, featuring supramolecular self-assembly in both the gel and the dielectric elastomer, enabled us to devise iontronic adhesives that outperform existing electroadhesives in mechanical compliance, resilience, conductor-dielectric interfacial toughness, and autonomous self-healing capabilities. The unique anchoring effect at the gel-elastomer contact prompted the construction of multi-layered, monolithic iontronic-adhesive soft grippers, in which OHGel and SHPU not only formed mechanically reliable interfaces, but also coupled to transduce electrical energy into actuation and astrictive adhesion. Materials innovation also triggered versatile functionalities in the grippers as manifested by weightlifting, picking up fragile matters, manipulating objects of various surface roughness, and remotely sensing the location of targets. Moreover, the inkjet printing protocol enabled automated, scalable, and micron-precision OHGel patterning, which greatly facilitated fast prototyping of iontronic devices with a wide range of form factors. While iontronic-adhesive grippers of millimeter scale have been devised in this work, further miniaturization is possible and promising to offer novel utility in microelectromechanical systems and small-scale biomedical robots. Other device embodiments, e.g., anti-gravity terrestrial robots and human-machine haptic interfaces, may also benefit from the self-healability, large adhesion output, and exteroceptive sensing capability of our soft iontronic-adhesive technology.

## Methods
### Polymer synthesis
P(SPMA$_{0.5}$-r-MMA$_{0.5}$) was synthesized through atom transfer radical polymerization (ATRP); SHPU was synthesized through step-growth polymerization. Detailed synthetic routes are provided in Supplementary Methods.

### Characterization of P(SPMA$_{0.5}$-r-MMA$_{0.5}$)
$M_n$ and PDI of P(SPMA$_{0.5}$-r-MMA$_{0.5}$) were acquired by GPC. The polyelectrolyte was dissolved in DI water (3 mg mL$^{-1}$), filtered through a sterilized filter (0.22 μm pore size; Ministart PES, Sartorius), then injected into a column (PL aquagel-OH 40, Agilent) for sample resolving at a flow rate of 1 mL min$^{-1}$. A refractive index detector in the chromatography system (Agilent 1100 Series) was employed to sample the retention time. A standard kit (EasiVial PEG/PEO 2 mL, Agilent) was used to calibrate the relationship between retention time and molecular weight. The molecular structure of P(SPMA$_{0.5}$-r-MMA$_{0.5}$) was examined by $^1$H NMR spectroscopy (DPX 400, Bruker).

### Preparation and characterization of pre-OHGel ink
To prepare the pre-OHGel ink, 4 wt% glycerol was mixed with deionized water, then 8 wt% P(SPMA$_{0.5}$-r-MMA$_{0.5}$) was dissolved in the binary solvent with 70 °C water bath and magnetic stirring for 1 h. Finally, 0.02 wt% polyether-modified siloxane surfactant (BYK-348, BYK Additives & Instruments) was added into the solution at room temperature as surface tension modifier. The as-prepared ink was

filtered through polytetrafluoroethylene (PTFE) syringe filters (0.45 μm pore size; C2008S1, Ossila) to remove impurities. Trace amounts of toluidine blue (T3260, Sigma-Aldrich) could be dissolved in the pre-OHGel ink to enhance the visibility of printed OHGel. The apparent viscosity of the inks with different polyelectrolyte load was examined under ambient condition using a rheometer (MCR 501, Anton-Paar). Surface tension of the pre-OHGel inks with or without BYK-348 surfactant was measured by a micro-tensiometer (Ez-Pi plus, Kibron). DOD inkjet printing was performed by a Dimatix Materials Printer (DMP-2800, Fujifilm).

## Characterization OHGels

3D topography of the printed OHGel lines was examined by an optical confocal microscope (Smartproof5, ZEISS) with a z-axis sampling resolution of 0.1 μm. FTIR spectra of OHGels with different PE/GY blending ratio were recorded by ATR-FTIR spectroscopy (Frontier, Perkin Elmer) in the range of 4000–600 cm$^{-1}$ at a nominal resolution of 1 cm$^{-1}$. Mechanical properties were characterized by uniaxial tensile tests using a universal mechanical tester (Criterion Model 42, MTS). Time-dependent viscoelastic behavior was studied by oscillatory rheometry using a rheometer (MCR 501, Anton-Paar), where OHGel samples were placed in a plate-plate geometry (8 mm diameter, 1 mm gap) and subjected to a frequency sweep from 0.1 to 100 Hz with 1% strain. Storage modulus ($G'$) and loss modulus ($G''$) were measured as a function of shear frequency. The self-healing process of OHGel (PE10/GY5) was monitored using an optical microscope (BX53, Olympus). Resistance change of OHGel electrodes upon damage, healing, and stretching was recorded using a LCR meter (E4980A, Keysight Technologies) at a sampling frequency of 1 kHz. Ionic conductivity of OHGel was measured by electrochemical impedance spectroscopy (EIS) using a potentiostat (Autolab PGSTAT30, Metrohm). An OHGel sample was sandwiched between two pieces of steel electrodes with a dimension of width × length × thickness = 10 × 10 × 1.5 mm, and stabilized at a given temperature (using oven, fridge, and ice bath). AC voltage was then applied across the gel with a logarithm sweep (10$^5$ Hz to 10$^{-2}$ Hz, 10 mV amplitude) to obtain the impedance spectra (Nyquist plot).

## Characterization of SHPU

Chemical Molecular structure of SHPU was examined by $^1$H and $^{13}$C NMR spectroscopy (DPX 400, Bruker). The hierarchical H-bonding in SHPU was revealed by transmission mode FTIR spectroscopy (Frontier, Perkin Elmer) in the range of 4000–400 cm$^{-1}$ at a nominal resolution of 1 cm$^{-1}$. FTIR spectra of SHPU in the C = O stretching region were deconvoluted by baseline correction and Gauss Lorenz curve fitting using OriginPro 2018. Thermal stability and decomposition temperature of SHPU were investigated by TGA (Q500, TA Instruments; heating rate: 10 °C min$^{-1}$, temperature range: 25–700 °C, nitrogen atmosphere). $T_g$ and viscoelastic property of SHPU were characterized by DMA (Q800, TA Instruments) through temperature sweep (heating rate: 3 °C min$^{-1}$, temperature range: −80 to 100 °C, frequency: 1 Hz, nitrogen atmosphere). The morphology of phase separation in SHPU was investigated by SAXS (Nanoinxider, Xenocs) with an exposure time of 600 s. A Linkam stage was employed to perform temperature control from −60 °C to 80 °C. To analyze the average distance between hard domains, d-spacing was calculated using the Bragg equation $d = 2\pi/q_{max}$. Stress-strain behavior, cycling hysteresis, and self-healing efficiency of SHPU were characterized by uniaxial tensile tests (Criterion Model 42, MTS). SHPU samples with a thickness of ~0.5 mm were cut into a dumbbell shape (ASTM D638-14 type V), then clamped by a pair of pneumatic grippers to prevent sample sliding during tensile stretching (loading rate: 40 mm min$^{-1}$). SHPU toughness was calculated by integrating the area beneath a stress-strain curve. In cycling test, a SHPU sample was stretched up to 1000% strain then released back to its original length for 10 consecutive cycles without resting between the cycles. In self-healing test, SHPU samples were bisected in the middle of the dumbbell shape, rejoined and self-healed at room temperature for different period, then stretched until fracture.

## OHGel-SHPU interaction

T-peel test (ASTM D1876-08) was conducted using a universal mechanical tester (Criterion Model 42, MTS) to investigate the interfacial toughness between OHGel and different dielectric elastomers. SHPU, Sylgard 184, and VHB 4905 tape were prepared with a dimension of 60 × 10 × 0.5 mm. OHGel ink was drop casted onto the films in an area of 40 × 10 mm, then gelated autonomously at room temperature. Microporous nylon membrane (HNWP04700, Merck) was used as the backing for OHGel since it allows OHGel to infiltrate and thus form tough bonding. 3 M Scotch tape was used as the backing for SHPU to prevent unwanted elongation. The as prepared gel-elastomer bilayers were stored in ambient condition for 24 h to fully develop interfacial bonding. T-peel tests were performed with a cross-head speed of 50 mm min$^{-1}$. Confocal Raman spectroscopy (Alpha 300 R, 633 nm laser source, WiTec) was performed to unravel the interfacial bonding between OHGel and SHPU. The spectra were collected from 900 to 1600 cm$^{-1}$ Raman shift at a nominal resolution of 3 cm$^{-1}$ (Continuous-wave, room temperature). All spectra were averaged over 20 traces with 20 s acquisition time being used. The spin unrestricted DFT calculations were conducted using the DMol package in Material Studio. Electron exchange and correlation were performed based on the generalized gradient approximation, where the Perdew-Burke-Ernzerhof functional (GGA-PBE) and the double numeric plus polarization (DNP) are used as the basis set. The Grimme's scheme was also employed to describe the van der Waals (VDW) interactions with the global orbital cut-off radius set to be 5 Å. The convergence criteria for geometric optimization were set as follow: $1.0 \times 10^{-5}$ Ha on energy, $5.0 \times 10^{-3}$ Å on displacement, and $2.0 \times 10^{-3}$ Ha Å$^{-1}$ on gradients.

## Iontronic-adhesive soft gripper

Materials preparation and characterization: to prepare the nanoporous carbon composite, styrene-ethylene-butylene-styrene (SEBS, TuftecTM, H1052) was dissolved in toluene (anhydrous, 99.8%; 244511, Sigma-Aldrich) by stirring at 60 °C for 2 h to obtain a 20 wt% viscous solution. CNT and graphite particles were mixed with the SEBS-toluene solution at a weight ratio of 1:1:5, then thoroughly dispersed by magnetic stirring to form a carbon ink. Silica nanoparticles (non-porous, 150 nm; 904414, Sigma-Aldrich) were dispersed in ethanol (anhydrous, > 99.5%; 459836, Sigma-Aldrich) through ultra-sonication for spray coating. Field emission SEM (JEOL 7600 F) was employed to reveal the morphology of the carbon composite and the silica nanoparticles. SHPU solution could be blade-coated or spin-coated to obtain different film thickness, followed by oven curing at 90 °C for 12 h. A surface profiler (Alpha-Step D-500, KLA) was utilized to check the thickness of all SHPU films. Dielectric constant of SHPU, Sylgard 184, and VHB 4905 was measured using a precision impedance analyzer (4294 A, Agilent) from 40 Hz to 10$^6$ Hz. Dielectric strength of these elastomers was measured using a Hipot tester (Chroma 9056) with a DC potential ramping rate of 500 V s$^{-1}$.

Fabrication: the actuator beam (unimorph DEA) of an iontronic gripping unit was prepared by inkjet printing OHGel electrodes (7.4 × 29 mm$^2$) and stacking SHPU thin films (9 × 32.5 mm$^2$) according to the exploded view in Fig. 4a. A thin layer of silica nanoparticles (0.6 mg cm$^{-2}$) was spray coated onto the outer surface of the contact SHPU layer to eliminate the tackiness of SHPU. Four gripping units were then assembled onto a 3D printed holder to complete the gripper. The capacitance of all the functional dielectric capacitors ($C_{ac}$ and $C_{ef}$) and the EDL capacitors ($C_{EDL}$) were measured using a precision impedance analyzer (4294 A, Agilent) from 40 to 10$^5$ Hz.

Object handling: a high voltage amplifier (610E, Trek) with overload protection was used to supply DC voltage (0.2–1 kV) to the iontronic gripper. Vertical movement of the gripper was enabled and

controlled by a motorized test stand (ESM303, Mark 10). To investigate the gripper's rapid release from metallic surfaces, the gripper was activated and adhered to the metallic cube for 1 min, then deactivated by turning off the power source. The whole process was recorded using a camera and the time interval between power off and detachment was extracted from the video. To investigate the gripper's adaption to extreme temperatures, a Peltier device (CP455535H, CUI Devices) was employed to regulate the temperature of the metallic cube (cooled down to −10 °C and heated up to 80 °C, respectively). The gripping process was recorded by a camera and an infrared thermometer (Ti200, Fluke) simultaneously and the temperature information was directly read out from the software provided by Fluke.

Exteroceptive sensation: the end effector was interfaced with a capacitance readout circuitry that consists of a microcontroller (Arduino UNO), a capacitance-to-digital converter (CDC, AD7746, Analog Devices), and two LEDs as visual indicators (Supplementary Fig. 32). The CDC module measures the absolute capacitance ($C_{ef}$) of the end effector, converts the analog signal to digital data, then communicate the data with Arduino through $I^2C$ protocol. The firmware that controls the above process was programmed in the Arduino software by referring to the open source code: https://github.com/interactive-matter/Arduino−AD7746.

## Iontronic-adhesive patch

Fabrication: iontronic-adhesive patches were fabricated by inkjet printing the interdigitated OHGel electrodes onto a 20 × 20 mm SHPU supporting layer (100 μm thick), making EDL contact, and encapsulating the electrodes with a 20 × 20 mm SHPU contact layer (60 μm thick). After that, a thin layer of silica nanoparticles (0.6 mg cm$^{-2}$) was spray coated onto the outer surface of the contact SHPU layer to eliminate the tackiness of SHPU.

Measurement of iontronic adhesion force: surface texture and $S_a$ of the testing substrates are examined using an optical confocal microscope (Smartproof5, ZEISS) with a z-axis sampling resolution of 0.1 μm. Normal force generated by an iontronic adhesive patch was measured using the customized setup as illustrated in Supplementary Fig. 26a. Specifically, the patch was connected to a probe through a piece of buffering foam, which is soft and can ensure a conformal contact between the patch and the target substrate. The other end of the probe was connected to a force gauge (0.2 mN resolution, M7-025, Mark 10) for dynamic force recording. To initiate the test, a 0.5 N preload in z-axis was applied to press the patch seamlessly on the substrate. DC voltage (0.2–1 kV) was then supplied and held for 1 min, followed by pulling the patch straight up using a motorized test stand (ESM303, Mark 10). The peak force recorded at the detaching moment is regarded as the normal force. For the measurement of shear force, the setup was rotated by 90° so that the relative motion between the substrate and the patch would happen along z-axis (Supplementary Fig. 26c). A 3 M Scotch tape was employed as the backing of the patch. The shear force (friction) upon pulling was recorded, and the peak force value featuring the transition from static to kinetic friction is regarded as the maximum static friction being generated between the patch and the substrate.

Power consumption and self-healing performance: charging, discharging, and leakage current of the iontronic-adhesive patch on aluminum and glass were monitored by measuring the voltage drop across a shunt resistor (10 MΩ) using a multimeter system (DAQ6510, Keithley). The power consumption of the patch could be calculated by multiplying the leakage current to the supplied voltage. To evaluate the self-healability of the iontronic-adhesive patch, a cut was made through the SHPU and OHGel layers, then rejoined and allowed to self-heal at ambient condition for 24 h. The performance (adhesion pressure and leakage current) of the patch were evaluated before and after the damage-healing process for comparison.

## Reporting summary

Further information on research design is available in the Nature Portfolio Reporting Summary linked to this article.

## Data availability

The data that support the findings of this study are available within this Article and its Supplementary Information. Raw data necessary to reproduce the figures within this Article are available in figshare database under https://doi.org/10.6084/m9.figshare.21716516.v1

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

## Acknowledgements

This work was supported by the National Research Foundation, Prime Minister's Office, Singapore, under the Campus of Research Excellence and Technological Enterprise programme, Smart Grippers for Soft Robotics project (grant No. SGSR CREATE), and partly supported by the National Research Foundation of Korean (NRF), Korean government (MSIT), under Grant No. 2021R1A2C3004420.

## Author contributions

D.G., T.G., J.Lee, and P.S.L. conceived the idea and designed the research. T.G. synthesized SHPU. J.Lee synthesized the polyelectrolyte and prepared OHGel. D.G., T.G., and J.Lee characterized the materials properties. D.G. formulated the pre-OHGel ink and characterized its inkjet printability. J.Lv prepared the nanoporous carbon composite. Y.L. conducted a DFT calculation. J.-H.C. conducted Raman spectroscopic test. J.X. conducted SEM imaging. D.G. designed, fabricated, and characterized the iontronic-adhesive grippers and patches. D.G. and Y.L. carried out device photography and videography. D.G., T.G., J.Lee, and P.S.L. prepared the manuscript with input from all authors. T.P. and P.S.L. supervised the research.

## Competing interests

The authors declare no competing interests.
