## [Peer review file · Nature Communications]

REVIEWER COMMENTS

Reviewer #1 (Remarks to the Author):

This work reports a set of materials (hydrogel electrode and TPU dielectric) and a printing fabrication process for soft electroadhesion. The results seem interesting, the figures are clear and the demos of good quality. There is a major problem with novelty and contribution though. I disagree with the authors about their analysis of the state of the art (weaknesses of current materials for electroadhesion) and major claimed contributions (ionotronic adhesion mechanism, better performing materials for electroadhesion).

I can see potential in this work given the high-quality of the results presented, but the authors should first reshape their article clarifying their novelty compared to the state of the art. The presentation of the mechanism is misleading. The comparison of materials performance for electroadhesion is not looking at the right metrics (e.g., electrostatic energy density, shear pressure, load to weight ratio). The working style should also be improved as currently it makes it difficult to understand the content.

Printing of stretchable electroadhesives and self-healing seem novel and can have advantages in electroadhesion but are now only presented as marginal parts of the work. I suggest that the authors expand on these aspects of their work, clarify their analysis of the state of the art and present clear novelty and contribution.

The details of my comments are given below.

The authors claim in the abstract an ionotronic adhesion mechanism. However, the adhesion mechanism seems to me to be no different from conventional Electroadhesion based on interdigitated electrodes insulated by a dielectric polymer. The authors replaced metals or conductive polymers with hydrogels as electrodes. Since Electroadhesion is based on electrostatics, replacing metals or conductive elastomers with hydrogels as electrodes does not change the mechanism for electroadhesion. Charge transportation mechanism within the electrodes changes, but electroadhesion is based on forces between static charges accumulated on the surfaces of the electrodes, which do not change between an electronic conductor and a hydrogel.

Low voltage (1 V) Ionotronic adhesion has been demonstrated and is based on direct contact between two hydrogels, unlike what is shown in this work, which is interdigitated hydrogel electrodes insulated with an elastomer dielectric (see Kim, H.J., Paquin, L., Barney, C.W., So, S., Chen, B., Suo, Z., Crosby, A.J., Hayward, R.C. Low-Voltage Reversible Electroadhesion of Ionoelastomer Junctions. *Advanced Materials* 2020).

Using hydrogel electrodes and Tpu-based dielectrics could have other advantages. Yet I do not see these other advantages claimed in a convincing way nor supported in the article.

The authors state (Introduction, page 2) that

“Electroadhesives based on elastomeric components, such as silicone and stretchable electronic conductor (e.g. carbon-filler-percolated composite) feature improved geometrical adaptation, but at a cost of the low dielectric constant of silicones, questionable conductor-dielectric interfacial bonding, and potential mechanical/electrical failures (tearing, puncturing, loss of conductivity) under large deformation.”

I do not agree with this analysis for the following reasons.

First, the role of the dielectric constant in the performance of electroadhesion is questionable. It might help slightly reducing the voltage, which might or might not be critical depending on the application, but alone it does not lead to higher performance. 1 - 5 kV voltages can be easily provided by palm-sized battery driven power supplies, so I do not see any revolutionary advantage in reducing the voltage from 5 to 1 kV. Performances of electroadhesion (similarly to DEAs and any other electrostatic actuator/clutch) do not depend on the dielectric constant ϵ alone, but are rather related to the maximum electrostatic energy density of the dielectric material, which is ϵE_{BD}^2 , with E_{BD} being the breakdown field (see Hinchet, R., Shea, H., 2020. High Force Density Textile Electrostatic Clutch. Advanced Materials Technologies). When replacing a dielectric material for an electrostatic actuator or an electroadhesion device, the metric to be compared should be the maximum energy density ϵE_{BD}^2 , not only the dielectric constant ϵ . I recommend the authors to include this comparison in their article. Based on the reported data, the electroadhesion devices developed in this work use a TPU based dielectric (SHPU) with $\epsilon = 7$ and operate with $E = V/g = (1000 \text{ "V" })/(2 * 60 \mu\text{"m" }) = 8.3 \text{ "V/" } \mu\text{"m"}$. Silicone instead has typically $\epsilon = 2.7$ and can reach and exceed breakdown fields values of $E = 100 \text{ "V/" } \mu\text{"m"}$ (Sylgard 184, see Albuquerque, F.B., Shea, H., 2020. Smart Mater. Struct.).

Therefore, from these data one cannot conclude any evident advantage in electrostatic forces when replacing Silicone with SHPU since the factor 2.6 gained on the dielectric constant would come at the price of a loss in electric field of factor 12.

Beyond electrostatics, performance of electroadhesion is highly influenced by mechanical and surface properties (see Cacucciolo, V., Shea, H., Carbone, G., 2022. Peeling in electroadhesion soft grippers. Extreme Mechanics Letters). I recommend that the authors include a proper comparison of their materials and devices with the state of the art, using the most widely accepted metrics: electrostatic energy density, shear pressure, load to weight ratio.

Secondly, the interfacial bonding between silicone and commonly used conductive silicones (e.g., silicone + carbon black) is generally very strong. These devices are typically fabricated by casting

uncured conductive silicone on top of the silicone membrane, leading to a bonding strength comparable to bulk material (see Eddings, M.A., Johnson, M.A., Gale, B.K., 2008. Determining the optimal PDMS–PDMS bonding technique for microfluidic devices. *J. Micromech. Microeng.* 18, 067001.)

As for the potential mechanical/electrical failures at large deformation, silicone based electroadhesive grippers have been demonstrated stretching over 150% while holding over 1.5 kg of weight repeatedly without any failure problem (see Cacucciolo, V., Shintake, J., Shea, H., 2019. Delicate yet strong: Characterizing the electro-adhesion lifting force with a soft gripper, in: 2019 2nd IEEE International Conference on Soft Robotics (RoboSoft)).

Reviewer #2 (Remarks to the Author):

This work reports an iontronic adhesion mechanism using a gel-elastomer system consisting of an ionic organohydrogel (OHGel) as adhesive electrodes and resilient polyurethane (SHPU) as dielectric layers. An ionic soft gripper that seamlessly integrates actuation, adhesive grip, and external perception is designed through additive manufacturing. The idea of this work is novel, the results are technically sound and well-delivered with ample evidence, and the manuscript is clearly written. The following comments should be addressed before the recommendation for publication can be made.

1. On Page 2, “Existing electroadhesives are commonly operated under a few kilovolts to generate sufficient adhesive force, and have limited materials selection to fabricate with”. The authors may want to be careful about this argument. The electroadhesion between some materials can be achieved at potentials as low as ~1V (see the paper titled Low-Voltage Reversible Electrodeposition of Ionomer Junctions, DOI: 10.1002/adma.202000600)

2. On Page 3, the schematics in Fig. 1c are nicely drawn but remain confusing. First, why does the architecture design illustrated in Fig. 1c differ from that in Fig. 4a? Which is the actual design of the soft iontronic adhesive unit used in this paper? Second, the iontronic adhesion mechanism is not explained in the manuscript. Why is the interdigitated OHGel pattern necessary for iontronic adhesion? This should be made clear in the paper. Third, the actuating mechanism of the DEA should also be clarified. Why does the DEA bend in response to voltage bias? That’s not self-evident. Furthermore, for figure legend of Fig. 1c, the word “adhension” is a typo.

3. Page 4. The word DEA (dielectric actuator?) should be defined at its first usage.

4. On Page 5, the author mentioned that the optimal OHGel with a polyelectrolyte-glycerol ratio of 2:1 exhibits balanced deformability, elasticity, and self-healing efficiency. However, no quantitative evidence was provided to corroborate the self-healing behavior of the OHGel. It is better to provide stress-stretch curves of OHGels before and after healing for various periods of time, and the self-healing efficiency should be discussed.

5. On page 5, “An increasing glycerol-polyelectrolyte ratio promotes water retention (Supplementary Fig. 3)” and thereby softens OHGel due to the enhanced degree of hydration. The reviewer thinks that water retention refers to the hydrogel’s ability to retain its initial water content in open environments, whereas “water retention” in the above sentences refers to the degree of water absorption or hydration. The authors may also want to quantify the mass change (due to potential water evaporation) of the OHGel over a reasonable period of time to demonstrate the water retention capability of the material.

6. On pages 6 and 7, the authors mentioned fatigue, a word referring to the initiation and propagation of cracks in a material due to cyclic loading, which is apparently not the content discussed in this work.

7. On page 6, schematics of the self-healing mechanisms in OHGel and SHPU are provided in Fig. 2h but not explained at all. What are the primary mechanisms underpinning the self-healing behavior of SHPU and, especially, OHGel?

8. On Page 9, the authors use Timoshenko analysis to predict the bending curvature of the DEA at rest. However, the Timoshenko method is only accurate for linear-elastic material and small-strain situations. For the soft materials presented in this work, results obtained by Timoshenko analysis can only be a very rough estimation. The author should clarify this in the manuscript.

9. On Page 10, it is mentioned that the contact layer employs a hydrophobic coating on its outer surface to obviate the inherent tackiness of SHPU and enables self-cleaning by preventing the collection of contaminants. The word self-cleaning is a bit misleading. We call lotus leaf a self-cleaning material considering its unique nanoscopic architecture of the leaf surface, but we do not regard plastic bottles (e.g., bottled water) as self-cleaning just because of their non-sticky nature.

10. On Page 10, Fig. 4a might be further improved by numbering each layer, so that one can know that the top SHPU is the first layer, the biased OHGel is the second layer, the stretched SHPU is the third layer,, the bottom SHPU with fingertips OHGel pattern is the 6th layer. The reviewer had a hard time figuring out the meaning of Fig. 4a.

Reviewer #3 (Remarks to the Author):

In this manuscript, the authors design and synthesize a supramolecular gel-elastomer system and use the inkjet printing technique to fabricate a soft gripper, which is composed of a dielectric elastomer actuator and an electrostatic adhesion segment. The soft gripper exhibits excellent adaptation to various soft and deformable objects and a large payload up to ~670. It is interesting that the use of ionic gels as the electrodes leads to fast release of the gripper from the objects. Whereas individual performance of the materials and devices have been demonstrated to different extents in literature (e.g. (1) soft grippers with intrinsic electroadhesion based on DEA, J. Shintake, 2015, *Adv. Mater.*; (2) integrated 3D printing of gel-elastomer system with robust heterolayer adhesion, P. Zhang, 2022, *Nature Comm.*; (3) device-level healing, self-healing soft electronics, J. Kang, 2019, *Nature Electronics*; devices with intrinsic self-healing properties of all device components, M. Khatib, 2018, *Small.*), this work achieves the performances in an integrated device at the same time. The authors have investigated the properties of the materials and the device in detail and the experimental results can support the main conclusions.

There are certain statements arising concerns about general concepts and must be rephased. Some detailed comments are as follows.

1. On page 2, the statement that “ion transportation in a swollen polymeric network will not be interrupted even under hyperelastic deformation” is not true. Depending on the ion transport mechanism, the deformation of polymer network will play a key part, e.g. by narrowing the ion transport path or by dragging the tethered ions along with the polymer chains, etc.
2. In fig. 1d, the chemical structures of glycerol and water do not have annotations. For example, do the red sphere represent oxygen atom?
3. Response time should be provided if “fast release from foreign surface” is mentioned repeatedly.
4. The operating voltage of the gripper herein is also on the order of 1 kV, so it is not convincing to emphasize “reduced voltage input”.
5. Table 1 might include one more column to compare the payloads of different devices, now that the payload of this work is much higher than others.
6. On page 5, the statement “The incorporation of nonvolatile, hygroscopic glycerol avoided unwanted solvent loss and no sign of gel stiffening was observed over the course of this study.” should be revised. Glycerol can help retain water but solvent loss is inevitably in general for gel materials, e.g. in elevated temperature or removed by the foreign bodies during contact or in aqueous environment, etc. (Kim, et. al, 2020, *Science.*)

7. The plot of Fig. 2k is confusing. The x-axis is tensile strain and the y-axis is resistance, while the curve is $R/R_0 = \lambda^2$.

8. On page 8, the authors claim that they “deliver the first gel-elastomer system that forms strong and inherent interfacial bonding without requiring additional surface treatment or coupling agent.”, which is too ambitious. See, for example, P. Zhang, 2022, Nature Comm. and Q. Ge, 2021, Sci. Adv.

Point-by-point response to the reviewers' comments

D. Gao, G. Thangavel, J. Lee *et al.*

Corresponding author: Pooi See Lee, pslee@ntu.edu.sg

Reviewer #1

This work reports a set of materials (hydrogel electrode and TPU dielectric) and a printing fabrication process for soft electroadhesion. The results seem interesting, the figures are clear and the demos of good quality. There is a major problem with novelty and contribution though. I disagree with the authors about their analysis of the state of the art (weaknesses of current materials for electroadhesion) and major claimed contributions (ionotronic adhesion mechanism, better performing materials for electroadhesion).

I can see potential in this work given the high-quality of the results presented, but the authors should first reshape their article clarifying their novelty compared to the state of the art. The presentation of the mechanism is misleading. The comparison of materials performance for electroadhesion is not looking at the right metrics (e.g., electrostatic energy density, shear pressure, load to weight ratio). The working style should also be improved as currently it makes it difficult to understand the content.

Printing of stretchable electroadhesives and self-healing seem novel and can have advantages in electroadhesion but are now only presented as marginal parts of the work. I suggest that the authors expand on these aspects of their work, clarify their analysis of the state of the art and present clear novelty and contribution.

Comment #1-1

The authors claim in the abstract an ionotronic adhesion mechanism. However, the adhesion mechanism seems to me to be no different from conventional Electroadhesion based on interdigitated electrodes insulated by a dielectric polymer. The authors replaced metals or conductive polymers with hydrogels as electrodes. Since electroadhesion is based on electrostatics, replacing metals or conductive elastomers with hydrogels as electrodes does not change the

mechanism for electroadhesion. Charge transportation mechanism within the electrodes changes, but electroadhesion is based on forces between static charges accumulated on the surfaces of the electrodes, which do not change between an electronic conductor and a hydrogel. Low voltage (1 V) Ionotronic adhesion has been demonstrated and is based on direct contact between two hydrogels, unlike what is shown in this work, which is interdigitated hydrogel electrodes insulated with an elastomer dielectric (see Kim, H.J., Paquin, L., Barney, C.W., So, S., Chen, B., Suo, Z., Crosby, A.J., Hayward, R.C. Low-Voltage Reversible Electroadhesion of Ionoelastomer Junctions. *Advanced Materials* 2020).

Response: We agree with the reviewer that our iontronic adhesives share the same physical principle (electrostatic/Coulomb attraction) with electroadhesives. When ionic gel-based electrodes are biased, mobile ions in the gel would accumulate and mirror the induced charges/dipoles in the opposing surface to generate electrostatic adhesion. Herein, we recognize “electrostatic adhesion” as the underlying mechanism for our devices, and acknowledge that “iontronic adhesion” may not represent a *new mechanism*. Instead, we shall describe it as a *new configuration* of electrostatic adhesion that uses ionic gel as electrodes. The statements in **Abstract** and **Introduction (page 3, line 4)** are modified in accordance with the above reasoning. On this basis, we classify our devices as *iontronic adhesives* by following the naming convention established by Suo’s group since 2018 (*Nat. Rev. Mater.* 3, 125–142, 2018). In this review, the authors provided an overview of devices that use ionic hydrogels as stretchable, transparent electrodes, and classified them as the first generation of iontronic devices. The formation of electrical double layers (EDLs) at ionic-electronic contact represents the key feature of iontronics. The usage of ionic conductors also motivated scientific research in the engineering of strong hydrogel-dielectric elastomer bonding, and the improvement of electrochemical stability at EDLs. Ionic conductor-electroded DEA (*Science* 341, 984–7, 2013), pressure sensor (*Adv. Mater.* 26, 7608–14, 2014), and electroluminescence (*Adv. Mater.* 33, 2101396, 2021; *Adv. Mater.* 32, e2005545, 2020) etc. function under the same mechanism as their electronic counterparts, still they are defined as iontronic devices from the perspectives of materials usage and the ionic-electronic hybridized configuration, instead of the underlying physics. Moreover, the existence of EDLs have the potential to raise unique device functionalities, such as the boosted power output in triboelectric energy harvesters (*Adv. Mater.* 29, 1702181, 2017)). In this work, EDLs are considered as functional impedance that discharge rapidly (*Phys. Chem. Chem. Phys.* 15, 8692–9, 2013) under open-circuit condition to facilitate the adhesives’ detachment from a conductive surface. In light

of the above statements, we believe it is proper to name our devices as *iontronic adhesives* and *iontronic-adhesive grippers*, and to describe their underpinning mechanism as *electrostatic adhesion*. Our previously defined “iontronic adhesion” mechanism is no longer used in the manuscript.

The reviewer also mentioned another work (*Advanced Materials* 32, 2000600, 2020) that realized reversible iontronic adhesion with ± 1 V. Under forwards bias, interfacial adhesion could be established between two ionoelastomers whose polymer chains are oppositely charged, through the formation of an ionic double layer. It’s obvious that such adhesion force can only form between the designated active materials; replacing one of the ionoelastomer to another material would fail to render the effect. Therefore, this work reported an iontronic adhesion phenomenon at materials level; it does not offer a universal solution to generate electrostatic adhesive force on an arbitrary foreign object, and it is not related to the devising of controllable adhesive modules for soft robotics. In contrast, our work focuses on the delivery of device-level iontronic adhesion, wherein the charged ionic gel electrodes can polarize most foreign passive surfaces/objects to render normal/shear adhesion force and perform gripping/handling functionalities. Therefore, it is fair to compare our work with the precedent device-level electrostatic adhesives as summarized in Table 1, whereas the ionoelastomer work represents the progress in another subcategory of electrostatic adhesion. To clarify this point, we have modified our statement on **Page 2, line 10** to “Existing device-level electroadhesives...”.

The authors state (Introduction, page 2) that “Electroadhesives based on elastomeric components, such as silicone and stretchable electronic conductor (e.g. carbon-filler-percolated composite) feature improved geometrical adaptation, but at a cost of the low dielectric constant of silicones, questionable conductor-dielectric interfacial bonding, and potential mechanical/electrical failures (tearing, puncturing, loss of conductivity) under large deformation.” I do not agree with this analysis for the following reasons.

Comment #1-2

First, the role of the dielectric constant in the performance of electroadhesion is questionable. It might help slightly reducing the voltage, which might or might not be critical depending on the

application, but alone it does not lead to higher performance. 1 - 5 kV voltages can be easily provided by palm-sized battery driven power supplies, so I do not see any revolutionary advantage in reducing the voltage from 5 to 1 kV. Performances of electroadhesion (similarly to DEAs and any other electrostatic actuator/clutch) do not depend on the dielectric constant ϵ alone, but are rather related to the maximum electrostatic energy density of the dielectric material, which is ϵE_{BD}^2 , with E_{BD} being the breakdown field (see Hinchet, R., Shea, H., 2020. High Force Density Textile Electrostatic Clutch. Advanced Materials Technologies). When replacing a dielectric material for an electrostatic actuator or an electroadhesion device, the metric to be compared should be the maximum energy density ϵE_{BD}^2 , not only the dielectric constant ϵ . I recommend the authors to include this comparison in their article. Based on the reported data, the electroadhesion devices developed in this work use a TPU based dielectric (SHPU) with $\epsilon = 7$ and operate with $E=V/g = (1000 \text{ "V"})/(2* 60 \mu\text{"m"}) = 8.3 \text{ "V/" } \mu\text{"m"}$. Silicone instead has typically $\epsilon = 2.7$ and can reach and exceed breakdown fields values of $E=100 \text{ "V/" } \mu\text{"m"}$ (Sylgard 184, see Albuquerque, F.B., Shea, H., 2020. Smart Mater. Struct.). Therefore, from these data one cannot conclude any evident advantage in electrostatic forces when replacing Silicone with SHPU since the factor 2.6 gained on the dielectric constant would come at the price of a loss in electric field of factor 12. Beyond electrostatics, performance of electroadhesion is highly influenced by mechanical and surface properties (see Cacucciolo, V., Shea, H., Carbone, G., 2022. Peeling in electroadhesion soft grippers. Extreme Mechanics Letters). I recommend that the authors include a proper comparison of their materials and devices with the state of the art, using the most widely accepted metrics: electrostatic energy density, shear pressure, load to weight ratio.

Response:

1) The significance of voltage reduction:

We employed 1 kV as the testing voltage to benchmark the adhesive patches' performance on different foreign substrates (**Fig. 5 c-d**). However, for many usage scenarios (like when handling metals (**Fig. 4d**) and smooth/high-k dielectrics), the adhesive patches can function under several hundred volts to generate sufficient shear pressures. Under 400 V input, the iontronic adhesive produces a shear pressure of 2.26 kPa, 1.28 kPa, and 0.50 kPa on aluminum, smooth glass, and grounded glass, respectively, which corresponds to a payload of 86.1 g, 50.6 g, and 19.8 g for our four-fingered iontronic-adhesive gripper (**Supplementary Fig. 26**). Decreasing the operating

voltage from kilovolts to hundred volts is a key step to enable small-scale, untether soft robots. While stepping up a battery output to kilovolts requires palm-sized power supplies, there is a vast selection of PCB-compatible, miniature, and ultralight ($< 1\text{g}$) power sources that can upscale a battery output up to 500 V. For instance, Ji *et al.* (Sci. Robot. 4, eaaz6451 (2019)) developed a 780 mg power board (including battery, transformer, sensor, controller) to transform a 3.7 V battery output to 450 V to drive low-voltage DEAs for robotic-insect locomotion. Moreover, the reduction in operating voltage is crucial for safe human-robot interaction.

Supplementary Fig. 26 | Maximum payload and shear pressure of the 4-fingered ionic-adhesive soft gripper on different materials under a series of voltage input.

2) Electrostatic energy density:

We agree with the reviewer that the peak performance of electrostatic adhesion is determined by the maximum energy density (κE_B^2) of the contact dielectric layer, where both dielectric constant (κ) and dielectric breakdown field (E_B) are deciding factors. However, in real world applications, electrostatic adhesives should not be operated near the marginal E_B for safety and reliability reasons. The advantage of having a high dielectric constant lies in that one can achieve the same level of electrostatic energy density, or adhesion performance, using a lower voltage input in the dielectric's safe range. By replacing Sylgard 184 ($\kappa \approx 2.8$) with SHPU ($\kappa \approx 6.8$), a 1.56 times reduction in operating voltage can be realized while rendering the same electrostatic energy density. Besides, 1 kV is a safe operating voltage for our devices rather than the breakdown voltage of SHPU. To calculate the maximum energy density of SHPU, we measured its dielectric strength by

following ASTM-D3755. As shown in **Supplementary Fig. 13a**, a SHPU specimen (thickness $\approx 200 \mu\text{m}$) was mounted between two opposing brass probes (diameter = 25 mm) with the testing cup filled with silicone oil to prevent flashover and premature breakdown. A Hipot tester (Chroma 9056) was employed to supply DC potential at a ramping rate of 500 V s^{-1} . The breakdown voltage (U_B) and SHPU film thickness (d) were recorded to calculate breakdown field ($E_B = U_B/d$). The results were analyzed by fitting into the two parameter Weibull distribution function:

$$P(E, \beta) = 1 - e^{-(E/E_B)^\beta}$$

where P is the cumulative probability of dielectric breakdown occurring at electric field equal to or below E , β is the shape parameter that describes the scattering of data, and E_B is the characteristic breakdown strength at a cumulative failure probability of 63.2%. To find E_B , the function was rearranged and plotted as $\log(-\ln(1-P))$ against E (Supplementary Fig. 13b). The characteristic dielectric strength of SHPU was determined to be $63.6 \text{ V } \mu\text{m}^{-1}$. This value is comparable with some commercial TPUs, such as Elastollan® 1185A10 (product of BASF, $E_B = 88.6 \text{ V } \mu\text{m}^{-1}$, *ACS Appl. Mater. Interfaces* 2017, 9, 6, 5237–5243). Based on the same method, the dielectric strength of VHB 4905 and Sylgard 184 (without prestretch) was measured to be $37.9 \text{ V } \mu\text{m}^{-1}$ and $27.2 \text{ V } \mu\text{m}^{-1}$ and, respectively.

Although the reviewer claimed that the dielectric strength of silicone can reach $100 \text{ V } \mu\text{m}^{-1}$ under $1.5 \times$ prestretch (*Smart Mater. Struct.* 29, 105024 (2020)), pristine Sylgard 184 (without prestretch) usually render a moderate dielectric strength as suggested by the Sylgard 184 Technical Data Sheet, Dow®. Because the dielectric contact layer in an electrostatic adhesive commonly endures none or little strain, the dielectric strength without prestretch should be used when calculating the energy density for electrostatic adhesion. Based on our calculation, the maximum electrostatic energy density for SHPU, VHB 4905, and Sylgard 184 are 0.243, 0.053, and 0.018 MJ m^{-3} , respectively. SHPU exhibits 12 times higher energy density than Sylgard 184.

Related document: Sylgard 184 Technical Data Sheet

<https://www.dow.com/content/dam/dcc/documents/en-us/productdatasheet/11/11-31/11-3184-sylgard-184-elastomer.pdf>

Supplementary Fig. 13 | Dielectric property of the elastomers. **a**, The setup for dielectric breakdown test. **b**, Weibull distribution of dielectric breakdown of SHPU, VHB 4905, and Sylgard 184. **c**, comparison of dielectric constant (100 Hz), dielectric strength, and maximum electrostatic energy density of SHPU, VHB 4905, and Sylgard 184.

3) Proper comparison of materials and devices:

Electrostatic energy density: The dielectric constant (κ), dielectric strength (E_B), and maximum electrostatic energy density of SHPU, Sylgard 184, and VHB 4905 are compared using bar charts as summarized in **Supplementary Fig. 13c**.

Shear pressure: We have summarized this parameter in **Table 1**, with operating voltage and target substrate being annotated (e.g. 1.0 kV | 1.3kPa, glass).

Payload-to-weight ratio: for electrostatic-adhesive soft grippers, only a few works (e.g. *Adv. Mater.* 28, 231–8 (2016); *RoboSoft* 108–113 (2019); *IEEE Trans. Ind. Electron.* 69, 642–651 (2022)) reported both the own weight and the payload of a specific gripper. We can thereby only perform a less comprehensive comparison on this parameter. We have summarized the available information from the above-mentioned literatures in **Supplementary Table 3**.

Supplementary Table 3 | Payload comparison of the reported electrostatic-adhesive grippers.

*Intrinsic elastomeric adhesion (stickiness, or Van der Waals force) of the contact layer was involved during the measurement.

Reference (DOI)	Gripper's weight (n)	Payload (m)	Driving voltage	Payload ratio (m/n)
10.1002/adma.201504264	1.5 g	82.1 g (metallic can)	3.5 kV	54.7*
10.1109/ROBOSOF T.2019.8722706	1.5 g	16 N (acrylic geometry connected to a force sensor)	3.5 kV	1088.4*
10.1109/TIE.2021.3053887	6.2 g	625 g (glass bottle filled with metal screws)	7 kV	100.8
This work	0.32 g	215 g (metallic cube and weights)	1 kV	671.9

Comment #1-3

Secondly, the interfacial bonding between silicone and commonly used conductive silicones (e.g., silicone + carbon black) is generally very strong. These devices are typically fabricated by casting uncured conductive silicone on top of the silicone membrane, leading to a bonding strength comparable to bulk material (see Eddings, M.A., Johnson, M.A., Gale, B.K., 2008. Determining the optimal PDMS–PDMS bonding technique for microfluidic devices. *J. Micromech. Microeng.* 18, 067001.) As for the potential mechanical/electrical failures at large deformation, silicone based electroadhesive grippers have been demonstrated stretching over 150% while holding over 1.5 kg of weight repeatedly without any failure problem (see Cacucciolo, V., Shintake, J., Shea, H., 2019. Delicate yet strong: Characterizing the electro-adhesion lifting force with a soft gripper, in: 2019 2nd IEEE International Conference on Soft Robotics (RoboSoft)).

Response: We appreciate the reviewer pointing out that the interfacial bonding between silicone and silicone-based composite conductors can be very strong, and that silicone-based electroadhesives grippers can hold extremely high payload repetitively (> 1000 cycles) without failure. We have corrected our statement on **page 2, line 15** as “Electroadhesives based on elastomeric components, such as silicone and stretchable electronic conductor (e.g. carbon-filler-percolated composite) feature improved geometrical adaptation and conductor-dielectric integration, but are limited by the low dielectric constant of silicones, as well as the irreversible loss in adhesive functionality upon severed damages (e.g. cutting, tearing, and puncturing).”

Reviewer #2

This work reports an iontronic adhesion mechanism using a gel-elastomer system consisting of an ionic organohydrogel (OHGel) as adhesive electrodes and resilient polyurethane (SHPU) as dielectric layers. An ionic soft gripper that seamlessly integrates actuation, adhesive grip, and external perception is designed through additive manufacturing. The idea of this work is novel, the results are technically sound and well-delivered with ample evidence, and the manuscript is clearly written. The following comments should be addressed before the recommendation for publication can be made.

Comment #2-1

On Page 2, “Existing electroadhesives are commonly operated under a few kilovolts to generate sufficient adhesive force, and have limited materials selection to fabricate with”. The authors may want to be careful about this argument. The electroadhesion between some materials can be achieved at potentials as low as $\sim 1\text{V}$ (see the paper titled Low-Voltage Reversible Electroadhesion of Ionoelastomer Junctions, DOI: 10.1002/adma.202000600)

Response: We appreciate the reviewer reminding us the work (*Advanced Materials*, 32, 2000600, 2020) that realized reversible electroadhesion with $\pm 1\text{V}$. Under forwards bias, interfacial adhesion could be established between two ionoelastomers whose polymer chains are oppositely charged, through the formation of an ionic double layer. It’s obvious that such adhesion force can only form between the designated active materials; replacing one of the ionoelastomer to another material would fail to render the effect. Therefore, this work reported an iontronic adhesion phenomenon at materials level; it does not offer a universal solution to generate electrostatic adhesive force on an arbitrary foreign object, and it does not contribute to the devising of controllable adhesive modules for soft robotics. In contrast, our work focuses on the delivery of device-level iontronic adhesion, wherein the charged ionic gel electrodes can polarize most foreign passive surfaces/objects to render normal/shear adhesion force and perform gripping/handling functionalities. Therefore, it is fair to compare our work with the precedent device-level electrostatic adhesives as summarized in Table 1, whereas the ionoelastomer work represents the progress in another subcategory of electrostatic adhesion. To clarify this point, we have modified our statement on **Page 2, line 10** to “Existing *device-level* electroadhesives...”.

Comment #2-2

On Page 3, the schematics in Fig. 1c are nicely drawn but remain confusing. First, why does the architecture design illustrated in Fig. 1c differ from that in Fig. 4a? Which is the actual design of the soft iontronic adhesive unit used in this paper? Second, the iontronic adhesion mechanism is not explained in the manuscript. Why is the interdigitated OHGel pattern necessary for iontronic adhesion? This should be made clear in the paper. Third, the actuating mechanism of the DEA should also be clarified. Why does the DEA bend in response to voltage bias? That's not self-evident. Furthermore, for figure legend of Fig. 1c, the word "Adhension" Is A Typo.

Response: 1) We thank the reviewer for pointing out the inconsistency between Fig. 1c and Fig. 4a. **Fig. 1c** is a conceptual illustration for the iontronic device, whereas **Fig. 4a** represents the actual design of iontronic adhesive unit with geometrical precision. We have clarified it in the caption of **Fig. 1c** to avoid misleading the readers. 2) The mechanism of iontronic adhesion is briefly mentioned on **Page 10, line 26**. The mechanisms for iontronic adhesion, DEA, and bendable DEMES are elaborated in **Supplementary Fig.1** with illustrative figures and detailed explanation. 3) We thank the reviewer for pointing out the typo and it has been corrected.

Supplementary Fig. 1 | Mechanisms of electrostatic adhesion and actuation. a-c, Schematic illustrations showing the mechanisms for **a)** electrostatic adhesion on conductive and dielectric surfaces, **b)** the mechanism for electrostatic actuation, and **c)** the formation and bending mechanism of DEMES unimorph.

Comment #2-3

Page 4. The word DEA (dielectric actuator?) should be defined at its first usage.

Response: We have provided the full name (dielectric elastomer actuator) for DEA at its first usage (page 4, line 2).

Comment #2-4

On Page 5, the author mentioned that the optimal OHGel with a polyelectrolyte-glycerol ratio of 2:1 exhibits balanced deformability, elasticity, and self-healing efficiency. However, no quantitative evidence was provided to corroborate the self-healing behavior of the OHGel. It is better to provide stress-stretch curves of OHGels before and after healing for various periods of time, and the self-healing efficiency should be discussed.

Response: We have supplemented the comparison between different OHGel compositions to showcase the advantages of PE10/GY5 (Supplementary Fig. 7a). The mechanical self-healing behavior of PE10/GY5 was quantitatively recorded via uniaxial tensile test (Supplementary Fig. 7b). After bisecting, rejoining, and healing the sample at room temperature for 15 minutes, it could restore 96.5% stretchability and 100% ultimate tensile strength.

Supplementary Fig. 7 | a, Comparison of OHGels with different PE/GY ratio. PE10/GY3 is free-standing yet barely heals due to the limited chain motion; PE/GY7 heals rapidly, but at a cost of becoming non-free-standing; PE10/GY5 occupies both good mechanical integrity and rapid self-healing capability. **b**, Mechanical self-healing behavior of the PE10/GY5 OHGel characterized by uniaxial tensile test. After bisecting, rejoining, and healing at room temperature for 15 minutes, the OHGel (PE10/GY5) could restore 96.5% stretchability and 100% ultimate tensile strength.

Comment #2-5

On page 5, “An increasing glycerol-polyelectrolyte ratio promotes water retention (Supplementary Fig. 3)” and thereby softens OHGel due to the enhanced degree of hydration. The reviewer thinks that water retention refers to the hydrogel’s ability to retain its initial water content in open environments, whereas “water retention” in the above sentences refers to the degree of water absorption or hydration. The authors may also want to quantify the mass change (due to potential water evaporation) of the OHGel over a reasonable period of time to demonstrate the water retention capability of the material.

Response: our statement on **page 5, line 14** should agree with the reviewer’s point that OHGel can preserve its initial water content, rather than continuously absorbing water from air. We have modified the statement as “An increasing glycerol-polyelectrolyte ratio preserves more initial water in OHGel and thereby softens OHGel due to the enhanced degree of polyelectrolyte-solvent interaction” to address this point. After inkjet printing the pre-OHGel ink (with high water content) on to a SHPU substrate, the formulation would partially dehydrate and then gelate physically under ambient condition. After ~ 6 hours, the OHGel samples could reach dynamic equilibrium in water evaporation/reabsorption, on condition that the ambient humidity is stable ($RH \approx 60\%$). Accordingly, an initial weight loss (**Supplementary Fig. 4a**) was observed for all OHGel samples with different PE/GY ratio. We further recorded their net weight for up to 14 days (room temperature, ambient air) and found no noticeable mass change in PE10/GY7 and PE10/GY5. PE10/GY3 further lost ~ 3.6% in net weight on day 14 due to its low glycerol content (**Supplementary Fig. 4b**).

g

Supplementary Fig. 4 | a, Net weight change in OHGels during the partial dehydration and gelation process after inkjet printing. Weight ratios are calculated by dividing measured weight by initial weight. **b**, Net weight change in OHGels recorded up to 14 days (ambient condition, RH \approx 60%). No noticeable mass change in PE10/GY7 and PE10/GY5 was observed. The slight fluctuation in weight could be due to the fluctuation in relative humidity. PE10/GY3 further lost \sim 3.6 wt% on day 14 due to its lower glycerol content. **c**, Weight fraction of water and glycerol in OHGels (in equilibrium with atmospheric moisture, RH \approx 60%).

Comment #2-6

On pages 6 and 7, the authors mentioned fatigue, a word referring to the initiation and propagation of cracks in a material due to cyclic loading, which is apparently not the content discussed in this work.

Response: We appreciate the reviewer pointing out that fatigue refers to the irreversible mechanical property change or fracture in a bulk material after being exposed to cyclic load. We agree that such phenomenon is not discussed in this work. The statement on **page 6, line 4** has been modified as “...that protects OHGel electrodes from mechanical damages such as accidental perforation”. The statement on **page 7, line 22** has been updated as “SHPU could recover from viscoelastic deformation as suggested by the almost identical stress-strain loops”.

Comment #2-7

On page 6, schematics of the self-healing mechanisms in OHGel and SHPU are provided in Fig. 2h but not explained at all. What are the primary mechanisms underpinning the self-healing behavior of SHPU and, especially, OHGel?

Response: The self-healing mechanisms for OHGel and SHPU are briefly mentioned on **page 8, line 4 and line 11**, respectively. A detailed discussion on OHGel self-healing mechanism is provided below **Supplementary Fig. 7**.

Self-healing mechanism of SHPU:

Based on the macromolecular design of SHPU, UPy motifs carrying quadruple H-bonding sites are introduced within its soft matrix to form dynamic crosslinks. UPy undergoes rapid association and dissociation of the quadruple H-bonding units (Fig. 2d, bottom) with a UPy-UPy dimer lifetime of 1.7 s. When a bulk SHPU sample is bisected and rejoined, dynamic chain motion/diffusion in the soft domains offers the opportunity for a pair of free, non-associated UPy groups from both halves to dimerize and reform dynamic crosslink at the cut. Since the T_g of soft domain is subzero (-12 °C), the polymer chain in soft domains exhibits high mobility at room temperature and can thus facilitate SHPU self-healing under ambient condition.

Self-healing mechanism of OHGel:

OHGel is formed by gelating P(SPMA_{0.5}-r-MMA_{0.5}) in a water/glycerol binary solvent. The amphiphilic polyelectrolyte comprises hydrophilic segments (SPMA) that dissolve in the polar solvent, and hydrophobic segments (MMA) that associate with each other to minimize their exposure to water. Such **hydrophobic interaction** gives rise to reversible physical crosslinks, with hydrophobes associating and disengaging in a dynamic equilibrium. The electromechanical self-healing of OHGel takes two steps. First, the solvent in bisected OHGel samples join immediately upon physical contact and thus form a connected pathway for ion transportation. Secondly, the free hydrophobes on the cut surfaces can reassociate, with the help of polymer chain diffusion, to regain mechanical strength at the cut. The autonomous self-healing efficiency of OHGel is determined by the crosslinking kinetics. A short crosslink lifetime and a high disassociation/reassociation rate is considered favorable for fast self-healing.

Comment #2-8

On Page 9, the authors use Timoshenko analysis to predict the bending curvature of the DEA at rest. However, the Timoshenko method is only accurate for linear-elastic material and small-strain situations. For the soft materials presented in this work, results obtained by Timoshenko analysis can only be a very rough estimation. The author should clarify this in the manuscript.

Response: We agree with the reviewer that Timoshenko analysis is accurate for linear elastic materials with small-strain deformation. The strain in SHPU is considerably small when forming the curved unimorph structure. Considering a total bimorph thickness of $\sim 260 \mu\text{m}$ and a maximum curvature of 1 cm^{-1} , the maximum strain endured by SHPU would be $130 \mu\text{m} \times 1 \text{ cm}^{-1} = 0.013$, or 1.3%. The stress-strain behavior of SHPU could be considered as linear elastic within 1.3% strain. Therefore we believe that our estimation wouldn't deviate too much from the real situation. Nonetheless, we have updated our statement on **page 10, line 7** as “The DEA adopts a dielectric elastomer minimum energy structure (DEMES) whose bending curvature at rest is **approximated** using Timoshenko analysis”.

Comment #2-9

On Page 10, it is mentioned that the contact layer employs a hydrophobic coating on its outer surface to obviate the inherent tackiness of SHPU and enables self-cleaning by preventing the collection of contaminants. The word self-cleaning is a bit misleading. We call lotus leaf a self-cleaning material considering its unique nanoscopic architecture of the leaf surface, but we do not regard plastic bottles (e.g., bottled water) as self-cleaning just because of their non-sticky nature.

Response: The incorporation of hydrophobic coating (silica nanoparticle) on SHPU can drastically reduce its surface energy, as is evidenced by the increment in water contact angle (CA) from 86° to 149° (**Supplementary Fig. 20a**). The coating thereby manifests superhydrophobicity, one of the underlying mechanisms that contribute to self-cleaning effect. As demonstrated in **Supplementary Fig. 20b**, after directly contacting the dirt, the pristine SHPU film became contaminated whereas the one with superhydrophobic coating remained clean. In light of the above evidence, the superhydrophobic surface of coated SHPU is essentially different from a non-sticky plastic film.

Supplementary Fig. 20 | Self-cleaning capability of the SHPU contact layer with hydrophobic coating. a, Water contact angle on pristine SHPU and SHPU coated with silica nanoparticles. **b,** SHPU membranes before and after contacting the dust. The pristine SHPU collected a lot of dust due to its inherent tackiness, whereas the one with hydrophobic coating maintained clean.

Comment #2-10

On Page 10, Fig. 4a might be further improved by numbering each layer, so that one can know that the top SHPU is the first layer, the biased OHGel is the second layer, the stretched SHPU is the third layer,, the bottom SHPU with fingertips OHGel pattern is the 6th layer. The reviewer had a hard time figuring out the meaning of Fig. 4a.

Response: We have numbered the layers in **Fig. 4a**. The Figure caption has been updated accordingly to annotate each layer with its designated number.

Fig. 4a, Structural design of an iontronic-adhesive gripping unit consisting of OHGel electrodes and SHPU dielectric layers (left). In the DEA module, a pair of parallel-plate OHGel electrodes (2, 4) are separated by a prestretched SHPU membrane (3), then encapsulated by passive SHPU layers (1, 5) on both sides. The thickness of top (1), middle (3, pre-stretched), and bottom (5) SHPU layer is $60\ \mu\text{m}$, $60\ \mu\text{m}$, and $80\ \mu\text{m}$, respectively. The end effector module comprising interdigitated OHGel electrodes and a SHPU contact layer (6, thickness = $60\ \mu\text{m}$) is further attached below layer (5) to form the gripping unit.

Reviewer #3

In this manuscript, the authors design and synthesize a supramolecular gel-elastomer system and use the inkjet printing technique to fabricate a soft gripper, which is composed of a dielectric elastomer actuator and an electrostatic adhesion segment. The soft gripper exhibits excellent adaptation to various soft and deformable objects and a large payload up to ~670. It is interesting that the use of ionic gels as the electrodes leads to fast release of the gripper from the objects. Whereas individual performance of the materials and devices have been demonstrated to different extents in literature (e.g. (1) soft grippers with intrinsic electroadhesion based on DEA, J. Shintake, 2015, *Adv. Mater.*; (2) integrated 3D printing of gel-elastomer system with robust heterolayer adhesion, P. Zhang, 2022, *Nature Comm.*; (3) device-level healing, self-healing soft electronics, J. Kang, 2019, *Nature Electronics*; devices with intrinsic self-healing properties of all device components, M. Khatib, 2018, *Small.*), this work achieves the performances in an integrated device at the same time. The authors have investigated the properties of the materials and the device in detail and the experimental results can support the main conclusions.

There are certain statements arising concerns about general concepts and must be rephased. Some detailed comments are as follows.

Comment #3-1

On page 2, the statement that “ion transportation in a swollen polymeric network will not be interrupted even under hyperelastic deformation” is not true. Depending on the ion transport mechanism, the deformation of polymer network will play a key part, e.g. by narrowing the ion transport path or by dragging the tethered ions along with the polymer chains, etc.

Response: We agree with the reviewer that the deformation of ionic gels can influence the behavior of ion transportation, and hence their ionic resistivity will increase under hyperelastic stretch. Such phenomenon has been experimentally validated by Keplinger *et al.* (*Science* 341, 984–7, 2013) in the research of hydrogel-electroded DEAs, where they found the increment of ionic resistivity is related to the concentration of electrolyte (**Fig. R1**). For the ionic hydrogel dissolving 5.48 M NaCl, its resistivity increased slightly from 0.016 $\Omega\cdot\text{cm}$ to 0.034 $\Omega\cdot\text{cm}$ when the sample was elongated 6 times of its initial length (due to narrowed ion transportation pathway in the polymeric network, etc.). We have modified our statement on **page 2, line 25** as “ionic gels can maintain ion

transportation (with slight increment in resistivity) even under hyperelastic deformation” to clarify this point.

Fig. R1 | Resistivity change of hydrogels with different NaCl concentration as functions of stretch. Reproduced from (Science 341, 984–7, 2013) with permission granted.

Comment #3-2

In fig. 1d, the chemical structures of glycerol and water do not have annotations. For example, do the red sphere represent oxygen atom?

Response: We have annotated the atoms’ color in the caption of **Fig. 1e**.

Comment #3-3

Response time should be provided if “fast release from foreign surface” is mentioned repeatedly.

Response: For our iontronic gripper, the response (release) time from a metallic foreign surface is summarized in **Fig. 4g**. Under 1 kV voltage input, the adhesive patched could retract from a smooth ($S_a = 0.12 \mu\text{m}$) and a rough ($S_a = 0.76 \mu\text{m}$) metallic surface within 5.9 s and 3.1 s, respectively. Faster release was observed when lower voltage (a few hundreds volt) was supplied.

Comment #3-4

The operating voltage of the gripper herein is also on the order of 1 kV, so it is not convincing to emphasize “reduced voltage input”.

Response: We used 1 kV as the standard driving voltage to benchmark the adhesive patches’ performance on different foreign substrates (**Fig. 5 c-d**). For many usage scenarios (like when handling metals and smooth/high-k dielectrics), the gripper can function under several hundred volts (**Fig. 4d**) to generate sufficient shear pressures. Under 400 V voltage input, the iontronic adhesive can produce a shear pressure of 2.26 kPa, 1.28 kPa, and 0.50 kPa on aluminum, smooth glass, and grounded glass, respectively, which corresponds to a payload of 86.1 g, 50.6 g, and 19.8 g for our four-fingered iontronic-adhesive gripper (**Supplementary Fig. 23**). We have reorganized our statement on **page 14, line 1** to clarify that the usage of OHGel and SHPU has helped to reduce the driving voltage of iontronic adhesion.

Supplementary Fig. 23 | Maximum payload and shear pressure of the 4-fingered iontronic-adhesive soft gripper on different materials under a series of voltage input.

Comment #3-5

Table 1 might include one more column to compare the payloads of different devices, now that the payload of this work is much higher than others.

Response: For electrostatic-adhesive soft grippers, only a few works (e.g. *Adv. Mater.* 28, 231–8 (2016); *RoboSoft* 108–113 (2019); *IEEE Trans. Ind. Electron.* 69, 642–651 (2022)) reported both the self-weight and the payload of a specific gripper. We can thereby only perform a less

comprehensive comparison on this parameter. We have summarized the available information from the above-mentioned literatures in **Supplementary Table 3**.

Supplementary Table 3 | Payload comparison of the reported electrostatic-adhesive grippers.

*Intrinsic elastomeric adhesion (stickiness, or Van der Waals force) of the contact layer was involved during the measurement.

Reference (DOI)	Gripper's weight (n)	Load (m)	Driving voltage	Payload (m/n)
10.1002/adma.201504264	1.5 g	82.1 g (metallic can)	3.5 kV	54.7*
10.1109/ROBOSOF T.2019.8722706	1.5 g	16 N (acrylic geometry connected to a force sensor)	3.5 kV	1088.4*
10.1109/TIE.2021.3053887	6.2 g	625 g (glass bottle filled with metal screws)	7 kV	100.8
This work	0.32 g	215 g (metallic cube and weights)	1 kV	671.9

Comment #3-6

On page 5, the statement “The incorporation of nonvolatile, hygroscopic glycerol avoided unwanted solvent loss and no sign of gel stiffening was observed over the course of this study.” should be revised. Glycerol can help retain water but solvent loss is inevitably in general for gel materials, e.g. in elevated temperature or removed by the foreign bodies during contact or in aqueous environment, etc. (Kim, et. al, 2020, Science.)

Response: We appreciate the reviewer addressing the difference between “water retention” and “solvent loss prevention”. We agree that even for non-volatile solvents with very high boiling point (such as glycerol), they can evaporate slowly and finally cause solvent loss after a very long period of time (over the length of this study). In our iontronic devices, the OHGel electrodes are properly encapsulated by SHPU, therefore they are not likely to contact foreign bodies directly to cause solvent loss. We have revised the statement on **page 5, line 13** as “hygroscopic glycerol helped to retain water” to address this point.

Comment #3-7

The plot of Fig. 2k is confusing. The x-axis is tensile strain and the y-axis is resistance, while the curve is $R/R_0 = \lambda^2$.

Response: The expression of $R/R_0 = \lambda^2$ is theoretically correct to describe the stretch-induced resistance change in an ideally stretchable conductor. We have rearranged it into $R = R_0(1+\varepsilon)^2$ to properly mirror the variables with the axis labels, where $\lambda = 1+\varepsilon$, λ is tensile stretch (ratio between final and initial length), ε is tensile strain. The statement on **page 8, line 9** was adjusted accordingly.

Comment #3-8

On page 8, the authors claim that they “deliver the first gel-elastomer system that forms strong and inherent interfacial bonding without requiring additional surface treatment or coupling agent.”, which is too ambitious. See, for example, P. Zhang, 2022, Nature Comm. and Q. Ge, 2021, Sci. Adv.

Response: We appreciate the reviewer reminding us of the latest progress in gel-elastomer hybrids with inherent interfacial bonding. We have hence modified the statement on **page 8, line 28** to avoid the inaccurate description. The literature (Q. Ge et al, *Sci. Adv.*, 2021, 7(2), eaba4261) has been mentioned on **page 8, line 28** to supplement our discussion. However, we are unable to identify “P. Zhang, 2022, Nature Comm.” based on the provided information.

REVIEWER COMMENTS

Reviewer #1 (Remarks to the Author):

I appreciate very much the thorough replies by the authors to my comments and the new results and clarifications included in the paper. I agree with some of the authors replies while I disagree with others (Comment #1-2).

The quality of the paper is significantly improved in this second version, but there are still some critical points to be clarified. In particular, the advantage of reducing the operating voltage needs to be supported by more grasping experiments below 500 V.

While I believe the claim that the electroadhesion performance is improved in this work compared electroadhesives made of silicone elastomers should be removed as it disagrees with data in literature.

Please see my comments below.

Comment #1-1

I am fully satisfied with the authors reply. The added clarification in the naming of the mechanism now prevents confusion.

Comment #1-2

I agree with the authors that reducing the voltage to under 500 V is a game changer for the PCB size. However, if this is a key result for this paper, then it should be included in the main rather than in Supplementary Information. Most of the demos shown in the main of the paper now show about 1 kV of operation (Fig. 4).

Please include demos that operate under 500 V in the main to support the ≤ 500 V operation as a main contribution of this work.

I disagree with the authors that reducing voltage from kV to 500 V is important for the safety of the user. Both 5 kV and 500 V voltages can be lethal, unless they are current-limited, as it is the case for most power supplies. It is the current, and not the voltage, that creates harm to the human body. Please notice that our bodies commonly reach several kV of potential during dry winters. There is a thorough discussion on this topic in the experimental section of Ji, X., Liu, X., Cacucciolo, V., Civet, Y., Haitami, A.E., Cantin, S., Perriard, Y., Shea, H., 2020. Untethered Feel-Through Haptics Using 18- μ m Thick Dielectric

Elastomer Actuators. *Advanced Functional Materials* n/a, 2006639.
<https://doi.org/10.1002/adfm.202006639>

Also, I disagree with the authors about the fact that “The advantage of having a high dielectric constant lies in that one can achieve the same level of electrostatic energy density, or adhesion performance, using a lower voltage input in the dielectric’s safe range.”. Please notice that “The dielectric safe range” decreases together with the breakdown field E_{BD} . Assuming for example that one operates a device at 80% of breakdown field, this value is 80 – 90 V per micrometer for Sylgard 184 PDMS, versus 8 – 9 V per micrometer of the SHPU dielectric material presented in this work. This means that the electrostatic energy density, which is the key electrical metric for adhesion performance, is significantly lower for SHPU than for Sylgard 184 PDMS.

Many thanks to the authors for performing breakdown strength tests using a rigorous procedure. These results are very helpful. However, I am surprised with some of the results. For example, the estimated breakdown strength of Sylgard 184 measured by the authors (27.2 V) seems well below values reported in literature for the same material, even when unstretched. Values beyond 100 kV per micrometer are reported for cumulative failure probability of 63.2%. Please check for example figure 8, S184 [10:1] in Hajjesmaili, E., Clarke, D.R., 2021. Dielectric elastomer actuators. *Journal of Applied Physics* 129, 151102. <https://doi.org/10.1063/5.0043959>.

The authors mention the breakdown field value reported in Sylgard 184 TDS. This value is measured on bulk material. Breakdown strength of bulk material is always much lower than for thin films (check again Hajjesmaili 2021), so this the value in TDS should not be taken into account.

Please clarify the discrepancy between your experiments and values reported in literature for breakdown strength of PDMS. I guess that the presence of silicone oil modifies the breakdown strength of Sylgard pdsm, as it is known that silicone oil penetrates silicone rubber. I suggest to repeat the experiments without the silicone oil bath.

Many thanks for the comparison reported in Table 3. This is very helpful.

I do not understand the * in Supplementary Table 3 though. Please explain. For example in *RoboSoft* 108–113 (2019) the authors explicitly report to have used a paper coating on the acrylic to avoid dry adhesion by Van Der Waals forces.

Comment #1-3

I am fully satisfied with the authors reply.

Reviewer #2 (Remarks to the Author):

The authors have thoroughly and nicely addressed the reviewer's comments, with the manuscript and supplementary information carefully revised. Now the manuscript is appropriate for Nature Communications and the reviewer would like to recommend its publication in the journal.

Reviewer #3 (Remarks to the Author):

The authors have carefully addressed the review comments and the manuscript should be almost ready for acceptance. Below are two minor issues:

- 1) In the revised Fig. 2k, the exponent 2 should be outside the parentheses.
- 2) The title of "P. Zhang, 2022, Nature Comm." is "Integrated 3D printing of flexible electroluminescent devices and soft robots".

Point-by-point response to the reviewers' comments

D. Gao, G. Thangavel, J. Lee *et al.*

Corresponding author: Pooi See Lee, pslee@ntu.edu.sg

Reviewer #1

I appreciate very much the thorough replies by the authors to my comments and the new results and clarifications included in the paper. I agree with some of the authors replies while I disagree with others (Comment #1-2).

The quality of the paper is significantly improved in this second version, but there are still some critical points to be clarified. In particular, the advantage of reducing the operating voltage needs to be supported by more grasping experiments below 500 V.

While I believe the claim that the electroadhesion performance is improved in this work compared electroadhesives made of silicone elastomers should be removed as it disagrees with data in literature.

Comment #1-2

I agree with the authors that reducing the voltage to under 500 V is a game changer for the PCB size. However, if this is a key result for this paper, then it should be included in the main rather than in Supplementary Information. Most of the demos shown in the main of the paper now show about 1 kV of operation (Fig. 4). Please include demos that operate under 500 V in the main to support the ≤ 500 V operation as a main contribution of this work.

Response: 1) As shown in Fig. 5e and Supplementary Video 7, we have supplemented demos showing that a two-fingered iontronic-adhesive gripper can pick-up aluminum foil, silicon wafer, and glass under 300, 340, and 400 V operating voltage, respectively.

I disagree with the authors that reducing voltage from kV to 500 V is important for the safety of the user. Both 5 kV and 500 V voltages can be lethal, unless they are current-limited, as it is the case for most power supplies. It is the current, and not the voltage, that creates harm to the human body. Please notice that our bodies commonly reach several kV of potential during dry winters.

There is a thorough discussion on this topic in the experimental section of Ji, X., Liu, X., Cacucciolo, V., Civet, Y., Haitami, A.E., Cantin, S., Perriard, Y., Shea, H., 2020. Untethered Feel-Through Haptics Using 18- μm Thick Dielectric Elastomer Actuators. *Advanced Functional Materials* n/a, 2006639. <https://doi.org/10.1002/adfm.202006639>

Response: 2) We appreciate the reviewer pointing out that it's the current and not voltage that causes safety issues to human body. We have read the above paper thoroughly to obtain better understanding on this topic. Note that we are not claiming any safety-related advantages of our iontronic-adhesives in the manuscript.

Also, I disagree with the authors about the fact that “The advantage of having a high dielectric constant lies in that one can achieve the same level of electrostatic energy density, or adhesion performance, using a lower voltage input in the dielectric’s safe range.”. Please notice that “The dielectric safe range” decreases together with the breakdown field E_{BD} . Assuming for example that one operates a device at 80% of breakdown field, this value is 80 – 90 V per micrometer for Sylgard 184 PDMS, versus 8 – 9 V per micrometer of the SHPU dielectric material presented in this work. This means that the electrostatic energy density, which is the key electrical metric for adhesion performance, is significantly lower for SHPU than for Sylgard 184 PDMS.

Response: 3) The breakdown field of SHPU is measured to be $63.6 \text{ V } \mu\text{m}^{-1}$ rather than $8\text{-}9 \text{ V } \mu\text{m}^{-1}$. The breakdown field of Sylgard 184 PDMS is measured to be lower than SHPU by using the same testing protocol (Supplementary Fig. 13). Based on our data, we claim that the electrostatic energy density of SHPU is higher than that of PDMS.

Many thanks to the authors for performing breakdown strength tests using a rigorous procedure. These results are very helpful. However, I am surprised with some of the results. For example, the estimated breakdown strength of Sylgard 184 measured by the authors (27.2 V) seems well below values reported in literature for the same material, even when unstretched. Values beyond 100 kV per micrometer are reported for cumulative failure probability of 63.2%. Please check for example figure 8, S184 [10:1] in Hajiesmaili, E., Clarke, D.R., 2021. Dielectric elastomer actuators. *Journal of Applied Physics* 129, 151102. <https://doi.org/10.1063/5.0043959>.

The authors mention the breakdown field value reported in Sylgard 184 TDS. This value is measured on bulk material. Breakdown strength of bulk material is always much lower than for thin films (check again Hajiesmaili 2021), so this the value in TDS should not be taken into account. Please clarify the discrepancy between your experiments and values reported in literature for breakdown strength of PDMS. I guess that the presence of silicone oil modifies the breakdown strength of Sylgard pdsm, as it is known that silicone oil penetrates silicone rubber. I suggest to repeat the experiments without the silicone oil bath.

Response 4) As suggested by the reviewer and the mentioned review paper (*Hajiesmaili, 2021*), breakdown strength is not an intrinsic material property, but a measured parameter that depends on sample geometries and testing conditions. Generally, the breakdown strength of an elastomer sample decreases with increasing **sample thickness** ($E_B \propto t^{-0.5}$) and **area** under field (more defects).

The discrepancy between our Weibull distribution and the literature may be due to the different thickness, area, and testing conditions of Sylgard 184. Our Sylgard 184 specimens have a thickness of $\sim 200 \mu\text{m}$ and a testing area of 4.91 cm^2 between the flat brass probes (probe diameter = 2.5 cm). As for the $> 100 \text{ V}/\mu\text{m}$ breakdown strength of Sylgard 184 mentioned in the review paper (Fig. 8, *Hajiesmaili, 2021*), we are unable to identify the samples' geometrical information as neither **Ref. 72** (*Smart Mater.Struct. 24(10), 105025, 2015*) cited in the paragraph nor **M. Kolloosche** (**Ref. 77**, *Appl. Phys.Lett. 96(7), 071904, 2010*) acknowledged in the figure caption leads to the origin of Fig. 8 (recited here below as Fig. R1a). Sufficient experimental information is required for a fair comparison in geometry.

What we learned from the review paper (*Hajiesmaili, 2021*) is that the reported over $100 \text{ V}/\mu\text{m}$ E_B of Sylgard 184 is measured with a hemispherical electrode resting on a slab of elastomer with a constant load (Fig. R1b), which corresponds to the configuration of Hertz indentation. The contact area between the hemispherical electrode tip and the elastomer has been estimated to be smaller than 0.5 mm^2 (*International Journal of Smart and Nano Materials 6.4 (2015): 290-303; authored by M. Kolloosche*), which is ~ 1000 times smaller than the flat disc electrodes employed in our experiment (Fig. R1c). Given a n times larger testing area, the breakdown voltage at 63.2% cumulative probability would decrease by a factor of $n^{-1/\beta}$ (page 19, *Hajiesmaili, 2021*), where β is the Weibull modulus. With $n = 1000$ and $\beta = 9.4$ (derived from our Weibull fitting), a 52%

reduction in E_B can be estimated. Considering other influencing factors such as different sample thickness, environmental humidity, and different setting in cutoff current that defines a breakdown event (which is 0.1 mA in our case), the lower E_B of Sylgard 184 in our measurement is justifiable. Oil bath has been commonly used in dielectric breakdown test to prevent O₂ induced flashover (arc) and premature breakdown. It's considered beneficial for obtaining more accurate E_B values. The experiments were conducted immediately after placing the elastomer in the oil cup so the influence of silicone oil impregnation should be minimal.

To avoid any mislead/dispute, we have mentioned in our main manuscript (**page 7, line 29**) that the reported E_B values are consistently measured and are valid when referring to our testing protocol. Breakdown strength may vary when different testing protocols are applied.

Fig. R1. a) Fitting cumulative Weibull distribution function to the dielectric breakdown voltage measurements of multiple elastomers. Reproduced from *Hajiesmaili, 2021* with permission. b) Dielectric breakdown measurement setup using a spherical top electrode (used by *M. Kollosche*). c) Dielectric breakdown measurement setup using a pair of flat brass electrodes (used in our experiment).

Many thanks for the comparison reported in Table 3. This is very helpful. I do not understand the * in Supplementary Table 3 though. Please explain. For example in RoboSoft 108–113 (2019) the authors explicitly report to have used a paper coating on the acrylic to avoid dry adhesion by Van Der Waals forces.

Response: 5) We have modified our *statement as “A paper coating is applied on the acrylic to avoid dry adhesion (Van der Waals force).”

Reviewer #2

Response: we greatly appreciate the acknowledgement and recommendation from reviewer 2.

Reviewer #3

The authors have carefully addressed the review comments and the manuscript should be almost ready for acceptance. Below are two minor issues:

Comment #3-1

In the revised Fig. 2k, the exponent 2 should be outside the parentheses.

Response: we appreciate the reviewer pointing out the issue and the expression has been corrected.

Comment #3-2

The title of "P. Zhang, 2022, Nature Comm." is "Integrated 3D printing of flexible electroluminescent devices and soft robots".

Response: We thank the reviewer for providing the detailed literature information. We have mentioned this work on **page 9, line 1** to supplement our discussion.

REVIEWERS' COMMENTS

Reviewer #1 (Remarks to the Author):

The reviewer thanks the authors for having carefully addressed all the comments. The article is improved in clarity and contents, ready to be published.

Reviewer #3 (Remarks to the Author):

The authors have addressed my comments. I think the manuscript can be accepted now.